# Disorder-specific alterations of tactile sensitivity in neurodevelopmental disorders

Jason L. He[1,2,3], Ericka Wodka[4,5], Mark Tommerdahl[6], Richard A. E. Edden[1,2], Mark Mikkelsen [1,2], Stewart H. Mostofsky[4,5,7,8] & Nicolaas A. J. Puts [1,2,3 ✉]

Alterations of tactile processing have long been identified in autism spectrum disorders (ASD) and attention-deficit/hyperactivity disorder (ADHD). However, the extent to which these alterations are disorder-specific, rather than disorder-general, and how they relate to the core symptoms of each disorder, remains unclear. We measured and compared tactile detection, discrimination, and order judgment thresholds between a large sample of children with ASD, ADHD, ASD + ADHD combined and typically developing controls. The pattern of results suggested that while difficulties with tactile detection and order judgement were more common in children with ADHD, difficulties with tactile discrimination were more common in children with ASD. Interestingly, in our subsequent correlation analyses between tactile perception and disorder-specific clinical symptoms, tactile detection and order judgment correlated exclusively with the core symptoms of ADHD, while tactile discrimination correlated exclusively with the symptoms of ASD. When taken together, these results suggest that disorder-specific alterations of lower-level sensory processes exist and are specifically related to higher-level clinical symptoms of each disorder.

[1] Russell H. Morgan Department of Radiology and Radiological Science, The Johns Hopkins University School of Medicine, Baltimore, MD 21287, USA. [2] F. M. Kirby Research Center for Functional Brain Imaging, Kennedy Krieger Institute, Baltimore, MD 21287, USA. [3] Department of Forensic and Neurodevelopmental Sciences, Sackler Institute for Translational Neurodevelopment, Institute of Psychiatry, Psychology, and Neuroscience, King's College London, London SE5 8AB, UK. [4] Center for Autism and Related Disorders, Kennedy Krieger Institute, Baltimore, MD, USA. [5] Department of Psychiatry and Behavioral Sciences, The Johns Hopkins University School of Medicine, Baltimore, MD 21287, USA. [6] Department of Biomedical Engineering, University of North Carolina at Chapel Hill, Chapel Hill, NC 27514, USA. [7] Center for Neurodevelopmental and Imaging Research, Kennedy Krieger Institute, Baltimore, MD 21287, USA. [8] Department of Neurology, The Johns Hopkins University School of Medicine, Baltimore, MD 21287, USA. ✉email: nicolaas.puts@kcl.ac.uk

The role that tactile processing plays throughout the human life course cannot be understated. Indeed, touch is the first sense to develop prenatally[1], beginning as early as 8 weeks into gestation. Infants receiving more tactile-kinesthetic stimulation from their caregivers are found to have better developmental outcomes, including greater weight gain, healthier sleep/wake cycles and accelerated motor development[2]. Over and above its effect on early physical development, tactile processing plays a critical role in social development[3]. Interpersonal "social touch" has a wide range of functions in early life[4]. Touch is critical to the formation of mother–infant attachments[5], a relationship that provides one of the initial platforms from which gesture-based[6] and verbal communication skills are first practiced and developed[7]. In early childhood, tactile perception affects the willingness of children to participate in rough-and-tumble play with peers[8], an act which is indispensable to early social development[8,9]. Of course, in later adolescence and adulthood, tactile processing also becomes an important factor in the development of romantic interpersonal relationships[10].

While tactile processing abnormalities have been reported in both ASD and ADHD[11,12], the extent to which these abnormalities are specific to each disorder (as opposed to being generally impaired across both disorders), and how they relate to the core behavioral phenotypes of each disorder remain unclear. Indeed, this latter question has recently been highlighted as an outstanding question for further inquiry by many in the field[12,13]. Although earlier versions of the diagnostic statistical manual of mental disorders (DSM) prohibited the co-diagnosis of ASD and ADHD (ASD + ADHD), the publication of the DSM-5[14] has since permitted a clinical diagnosis of ASD + ADHD. This change in diagnostic practice has implications for how we characterize and treat each of these disorders, bringing forward the need for more formal investigations into the degree to which previously identified symptoms are specific and/or general to each disorder. In the context of sensory processing abnormalities in ASD and ADHD, it is possible that some or even all of the previously identified sensory abnormalities in one disorder were simply due to the pathophysiology of the other. Indeed, while elevated detection thresholds and wider temporal binding windows have been identified in ASD across a range of sensory domains[15–19], findings have been mixed[17,20–26], with many studies not controlling for ADHD symptomatology. Problematically, elevated detection thresholds and wider temporal binding windows have also been identified in ADHD, suggesting that the similar findings in ASD may actually be explained by co-occurring ADHD. Additionally, it is possible that sensory abnormalities that would not otherwise be present in isolated cases of either disorder are only present in individuals with a co-diagnosis of ASD + ADHD. Indeed, having ASD + ADHD could even mediate or moderate any existing sensory sensitivities specific to each disorder in isolation.

Thus far, studies which have aimed to characterize tactile processing abnormalities in ASD and ADHD have typically relied on subjective self/caregiver report questionnaires[11,27–30]. While informative in their own right[28], these are limited in their ability to address the disorder-specificity of sensory and, more specifically, tactile abnormalities in ASD and ADHD, as well as whether and how they relate to the core symptoms of each disorder. For instance, many of the embedded questions assess an individual's subjective emotional reaction to sensory stimuli (e.g., hyper-responsivity and hypo-responsivity to tactile stimuli), rather than provide an objective assessment of actual alterations in sensory processing. Moreover, while many questionnaires differentiate between sensory domains (e.g., sight, touch, and hearing), they collectively do not differentiate between perceptual domains (e.g., detection, discrimination, and judgement). This latter limitation is perhaps the most important.

By failing to differentiate between perceptual domains, which represent distinct aspects of cortical processing, questionnaires can only capture sensory abnormalities in broad brush strokes and likely lack the resolution to distinguish disorder-specific alterations in the cortical processing of sensory information.

Through psychophysics, it is possible to objectively quantify individual differences in sensory processing across perceptual domains and subdomains[31]. With the aim to determine the extent to which abnormalities in tactile processing are specific to ASD and ADHD, we compared psychophysically derived tactile perceptual thresholds across a large sample of children (8–12 years of age) with ASD with no co-diagnosis of ADHD ($n = 34$), ADHD with no co-diagnosis of ASD ($n = 104$), combined ASD + ADHD ($n = 99$) and typically developing controls ($n = 197$; see Table 1 in "Methods" section for full details). Our analyses focused on three separate subdomains of tactile processing, namely, detection, discrimination and order judgement. These subdomains were assessed using a battery of protocols. Importantly, these protocols were specifically selected as they provide an assessment of the integrity of GABAergic cortical processes that have long been suggested to underlie the sensory perceptual abnormalities that have been identified in both ASD and ADHD.

Results from our initial group comparisons on the performance metrics from each protocol suggested that while all three groups showed alterations of tactile processing relative to typically developing controls, these alterations were dependent on the subdomain being compared, highlighting the importance of differentiating between different subdomains of processing when investigating sensory alterations in these disorders. We then examined how sensitivity across subdomains related to reports of sensory-related and disorder-specific symptoms of each disorder. Across all participants, individuals who had higher (i.e., worse) detection thresholds were also those reported to have more problems with both hyper-responsivity and hypo-responsivity to sensory stimuli, as well as problems with social participation. When correlating measures from the tactile processing subdomains to the core clinical symptoms of each disorder, a remarkable pattern emerged. Our results showed that the disorder-specific tactile processing alterations identified in our initial group comparisons were also the metrics that were most commonly and specifically related to the core symptoms of each respective disorder. For instance, higher amplitude and frequency discrimination thresholds, which were more common to children with ASD (including ASD + ADHD), were exclusively related to problems with communication in ASD. Conversely, order judgement thresholds, which was more commonly elevated in children with ADHD (including ASD + ADHD), was exclusively related to problems with inattention in ADHD. As expounded below, our results provide an important description of tactile processing abnormalities in ASD and ADHD, showing that although children with ASD, ADHD, and ASD + ADHD do indeed have alterations in tactile processing, the perceptual domains in which those alterations occur, and how they relate to clinical symptomology, follow a disorder-specific pattern. The neurophysiological implications of our results are discussed, with specific reference to the potential cortical GABAergic processes that might be affected in these disorders.

## Results

It is possible to assess an individual's ability to perceive tactile stimuli through the application of adaptive two-alternative force choice psychophysical paradigms. These paradigms allow for a quick assessment of an individual's perceptual sensitivity by identifying their sensory thresholds. Sensory thresholds refer to the weakest stimulus input (or change relative to a standard stimulus) an individual can accurately perceive and are relatively stable characteristics across time[32]. Generally speaking, lower

**Table 1 Descriptive statistics for demographic based statistics by group.**

| | Control | | | ASD | | | ADHD | | | ASD + ADHD | | |
|---|---|---|---|---|---|---|---|---|---|---|---|---|
| | *n* | *M* | *SD* | *n* | *M* | *SD* | *n* | *M* | *SD* | *n* | *M* | *SD* |
| *General* | | | | | | | | | | | | |
| Age | 197 | 10.20 | 1.20 | 34 | 10.34 | 1.25 | 104 | 9.94 | 1.20 | 99 | 10.48 | 1.35 |
| Gender (Female) | 30 | – | – | 13 | – | – | 30 | – | – | 7 | – | – |
| Handedness (Left) | 20 | – | – | 13 | – | – | 12 | – | – | 6 | – | – |
| SES | 191 | 54.26 | 9.07 | 34 | 51.09 | 9.43 | 102 | 51.21 | 12.53 | 92 | 50.21 | 9.80 |
| *WISC 4* | | | | | | | | | | | | |
| Full-scale IQ | 139 | 115.85 | 12.27 | 18 | 110.11 | 14.55 | 75 | 107.87 | 10.98 | 56 | 97.63 | 14.70 |
| Perceptual reasoning | 139 | 112.27 | 12.61 | 18 | 114.06 | 10.43 | 75 | 109.13 | 11.12 | 56 | 103.21 | 14 |
| Processing speed | 139 | 102.80 | 13.06 | 18 | 94.22 | 14.29 | 75 | 95.57 | 12.42 | 56 | 85.84 | 17.67 |
| Verbal communication | 139 | 118.93 | 12.90 | 18 | 110.61 | 19.71 | 75 | 111.96 | 12.63 | 56 | 104.68 | 15.11 |
| Working memory | 139 | 111.65 | 14.91 | 18 | 108.56 | 14.36 | 75 | 102.61 | 12.82 | 56 | 91.66 | 15.37 |
| *WISC 5* | | | | | | | | | | | | |
| Full-scale IQ | 83 | 113.24 | 12.58 | 16 | 100.94 | 14.90 | 52 | 107.25 | 13.92 | 43 | 96.33 | 16.20 |
| Fluid reasoning index | 83 | 122.55 | 109.29 | 16 | 101.88 | 12.39 | 52 | 108.94 | 14.73 | 43 | 102.33 | 15.96 |
| Visual spatial index | 83 | 111.08 | 102.05 | 16 | 108.81 | 15.98 | 52 | 107.44 | 15.26 | 43 | 102.05 | 15.66 |
| Processing speed | 83 | 106.72 | 13.15 | 16 | 94.44 | 14.93 | 52 | 96.96 | 13.61 | 43 | 89.14 | 15.86 |
| Verbal communication | 83 | 111.70 | 10.77 | 16 | 100.44 | 15.39 | 52 | 108.40 | 13.62 | 43 | 98.77 | 16.19 |
| Working memory | 83 | 111.29 | 13.52 | 16 | 100.06 | 18.03 | 52 | 106.08 | 15.16 | 43 | 96.37 | 15.49 |
| *Conners: parent report* | | | | | | | | | | | | |
| Total inattention | 71 | 44.70 | 4.78 | 10 | 54.60 | 6.24 | 40 | 73.13 | 8.60 | 26 | 70.08 | 9.87 |
| Total hyperactive | 71 | 46.72 | 5.09 | 10 | 51.80 | 3.29 | 40 | 72.65 | 14.52 | 26 | 69.46 | 13.16 |
| *Conners 3: parent report* | | | | | | | | | | | | |
| Total inattention | 133 | 47.35 | 8.45 | 25 | 64.64 | 11.35 | 75 | 76.81 | 10.38 | 73 | 78.62 | 10.27 |
| Total hyperactive | 71 | 46.72 | 5.09 | 10 | 51.80 | 3.29 | 40 | 72.65 | 14.52 | 26 | 69.46 | 13.16 |
| *DuPaul's ADHD rating scale* | | | | | | | | | | | | |
| Total hyperactivity | 193 | 2.36 | 2.58 | 34 | 8.15 | 4.89 | 103 | 14.98 | 6.58 | 95 | 13.56 | 6.01 |
| Total inattention | 193 | 3.14 | 2.84 | 34 | 10.91 | 5.38 | 103 | 19.09 | 4.56 | 95 | 18.66 | 5.39 |
| *ADOS* | | | | | | | | | | | | |
| Total score | 0 | – | – | 6 | 16 | 2.28 | 0 | – | – | 13 | 15.85 | 2.94 |
| Communication and social score | 0 | – | – | 6 | 11.83 | 1.60 | 0 | – | – | 13 | 13.31 | 2.29 |
| Social interaction | 0 | – | – | 6 | 8.17 | 1.47 | 0 | – | – | 13 | 9.31 | 1.93 |
| Stereotyped behaviors | 0 | – | – | 6 | 4.17 | 2.23 | 0 | – | – | 13 | 2.54 | 1.27 |
| *ADOS 2* | | | | | | | | | | | | |
| Total score | 0 | – | – | 28 | 15.04 | 4.88 | 0 | – | – | 81 | 14.52 | 4.67 |
| Communication and social score | 0 | – | – | 28 | 3.50 | 1.48 | 0 | – | – | 85 | 3.56 | 1.56 |
| Social interaction | 0 | – | – | 28 | 8.32 | 2.67 | 0 | – | – | 81 | 8.22 | 3.09 |
| Stereotyped behaviors | 0 | – | – | 28 | 3.21 | 1.93 | 0 | – | – | 81 | 2.69 | 1.59 |
| *ADI-R* | | | | | | | | | | | | |
| Reciprocal social interaction | 0 | – | – | 22 | 20.27 | 5.23 | 0 | – | – | 68 | 20.99 | 5.69 |
| Communication | 0 | – | – | 22 | 16.45 | 3.79 | 0 | – | – | 69 | 16.55 | 4.58 |
| Restrictive repetitive/Stereotyped behavior | 0 | – | – | 22 | 6.14 | 1.75 | 0 | – | – | 69 | 6.25 | 2 |
| *SPM* | | | | | | | | | | | | |
| Social participation | 124 | 44.66 | 6.35 | 18 | 59.78 | 7.80 | 60 | 54.33 | 7.90 | 50 | 65.88 | 5.12 |
| Vision | 124 | 45.35 | 6.56 | 18 | 59.72 | 8.98 | 60 | 51.13 | 8.17 | 50 | 59.50 | 8.56 |
| Hearing | 124 | 46.02 | 5.80 | 18 | 58.72 | 8.68 | 60 | 51.77 | 8.98 | 50 | 66.56 | 6.02 |
| Touch | 124 | 44.95 | 6.54 | 18 | 57.94 | 8.71 | 60 | 51.07 | 8.62 | 50 | 63.94 | 7.98 |
| Total sensory systems | 124 | 44.13 | 5.63 | 18 | 58.39 | 7.91 | 60 | 52.87 | 7.77 | 50 | 64.52 | 5.27 |
| Body awareness | 124 | 43.05 | 5.98 | 18 | 54.61 | 9.43 | 60 | 53.43 | 9.05 | 50 | 61.26 | 7.22 |
| Balance motion | 124 | 44.82 | 6.39 | 18 | 53.72 | 7.89 | 60 | 51.30 | 8.16 | 50 | 60.42 | 7.33 |
| Planning ideas | 124 | 45.07 | 5.57 | 18 | 59.44 | 9.07 | 60 | 56.07 | 7.58 | 50 | 65.80 | 6.74 |
| *SEQ* | | | | | | | | | | | | |
| Total hypersensitivity | 50 | 41.18 | 6.55 | 16 | 73.31 | 14.97 | 6 | 49.33 | 10.58 | 39 | 86.18 | 20.24 |
| Total hyposensitivity | 50 | 21.58 | 3.69 | 16 | 28.94 | 6.92 | 6 | 24.50 | 6.44 | 39 | 37.41 | 9.44 |
| Total sensory seeking | 50 | 42.94 | 10.67 | 16 | 68.19 | 22.12 | 6 | 54.50 | 13.63 | 39 | 76.13 | 17.85 |
| Social hypersensitivity | 50 | 12.68 | 2.79 | 16 | 23.31 | 7.59 | 6 | 15.17 | 5.19 | 39 | 27.13 | 7.06 |
| Social hyposensitivity | 50 | 5.52 | 1.46 | 16 | 7.88 | 2.96 | 6 | 6.50 | 2.81 | 39 | 10.90 | 3.13 |
| Social sensory seeking | 50 | 7.40 | 1.95 | 16 | 10.62 | 3.59 | 6 | 10.33 | 2.88 | 39 | 11.85 | 3.69 |
| Non-social hypersensitivity | 50 | 25.70 | 4.20 | 16 | 44 | 8.35 | 6 | 30.17 | 7.05 | 39 | 50.46 | 14.89 |
| Non-social hyposensitivity | 50 | 16.06 | 2.75 | 16 | 21.06 | 4.71 | 6 | 18 | 4.15 | 39 | 26.51 | 7.46 |
| Non-social sensory seeking | 50 | 35.54 | 9.34 | 16 | 57.56 | 19.79 | 6 | 44.17 | 11.92 | 39 | 64.23 | 15.79 |

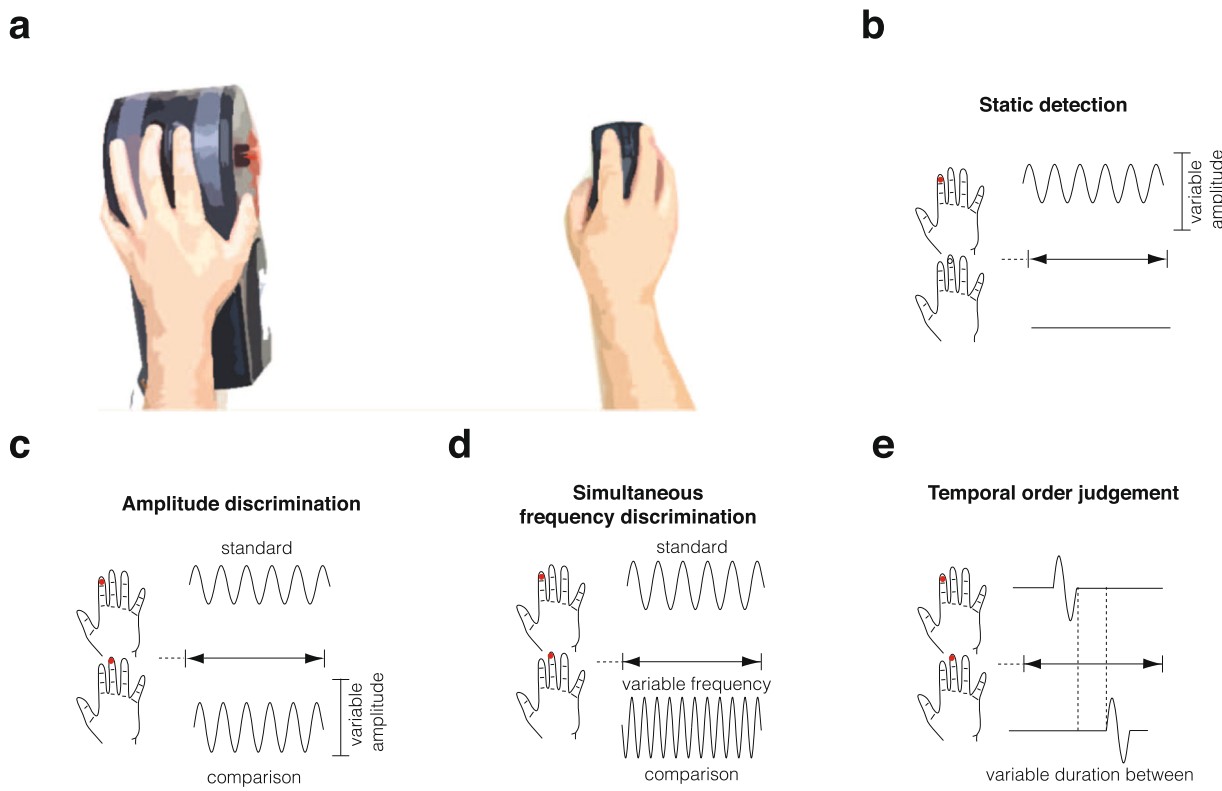

**Fig. 1 Overview of the setup and completed protocols. a** Participants placed their left hand on the vibrotactile stimulator and their right hand on a computer mouse. The stimulator delivered sinusoidal pulses to the left digit 2 and 3 via two 5 mm cylindrical probes. Responses were made with right digits 2 and 3 All stimuli were delivered between 0–350 μm and between 0–50 Hz. **b** Visual schematic of the static detection protocol. A single tactile stimulus was pseudorandomly delivered to either left digit 2 or left digit 3 of a child's left hand at the start of every trial. Following stimulation, participants were asked on which finger they believed they had received the stimulus and were asked to respond using the fingers of their opposite hand. An adaptive staircasing procedure was used to modify the trial difficulty. If the participant was able to accurately identify the correct site of stimulation, the amplitude of the stimulus in the subsequent trial would decrease, making it more difficult to detect, if the participant was unable to accurately identify the correct site of stimulation, the amplitude of the stimulus in the subsequent trial would increase, making it more easy to detect. **c** Visual schematic of the amplitude discrimination protocol. Participants received simultaneous stimulation of their left digit 2 and left digit 3 and were asked to report which digit they believed they had received the stimulus with the higher amplitude. In each trial, a standard stimulus was delivered to one finger while a more intense 'comparison stimulus' was delivered to the other. Localization was randomized between left digit 2 and left digit 3. Difference in amplitude between the standard stimulus and the comparison stimulus was adjusted based on participant performance so that the difference between the stimuli would decrease (making the stimuli harder to discriminate) with a correct responses or increase with incorrect responses (making it harder to discriminate). **d** Visual schematic of the frequency discrimination protocol. Two stimuli were delivered simultaneously to left digit 2 and left digit 3. Participants were asked which finger they believed had received the higher frequency stimulus. Adaptative staircasing was applied to the difference in frequency between the standard and comparison stimulus, increasing and decreasing when participants were correct or incorrect, respectively. **e** Visual schematic of the temporal order judgement protocol. Two brief vibrotactile pulses were delivered to left digit 2 and left digit 3 separated temporally by a starting intertrial interval of 150 ms (the first pulse was assigned to either digit pseudo-randomly). Participants were asked to make a response based on which digit they believed received the first pulse. The intertrial interval was increased or decreased by 10% for incorrect and correct responses, respectively. Other protocols (simple and choice reaction time, dynamic detection, sequential frequency discrimination, and temporal order judgement with a carrier stimulus) were also completed but are not presented in the main manuscript for focus and brevity. Further clarification of the setup and the protocols presented in this figure are provided in the "Methods" section, while information about the other protocols that are not presented here can be found in Supplementary Methods.

thresholds imply greater sensitivity, while higher thresholds imply worse sensitivity. Through a battery of these protocols (see Fig. 1a for an overview of the setup), we assessed static detection (Fig. 1b), sequential amplitude and simultaneous frequency discrimination (Fig. 1c, d), and order judgement thresholds (Fig. 1e) for tactile stimuli delivered to the left index (i.e., left digit 2) and middle finger (i.e., left digit 3) of children with ASD, ADHD, ASD + ADHD, and typically developing controls. An overview of the protocols can be seen in Fig. 1. Further information on the protocols presented here can be found in the "Methods" section, while details of additional protocols that were completed but not presented here can be found in Supplementary Methods, Supplementary Fig. 1.

**Detection**. Results from group comparisons on performance outcomes of the detection protocol are shown in Fig. 2. There was a significant main effect of group on detection thresholds, $(F(3, 375) = 11.24, p < 0.001; \eta^2_p = 0.08; BF_{10} = 3.49^{07})$. There was also a main effect of group on median reaction times $(F(3, 402) = 7.91, p < 0.001; \eta^2_p = 0.06; BF_{10} = 2.78^{12})$, accuracy $(F(3, 373) = 7.13, p = 0.001)$ and the number of reversals $(F(3, 421) = 14.31, p < 0.001; \eta^2_p = 0.06; BF_{10} = 689,653.94)$. The ADHD $(t(375) = 4.51, p_{Tukey} < 0.001; d = 0.60; BF_{10} = 2803.56)$ and ASD + ADHD $(t(375) = 4.83, p_{Tukey} < .001; d = 0.70; BF_{10} = 16,112.13)$ groups had significantly higher detection thresholds compared to the typically developing control group, while the ASD group did not $(t(375) = 1.62, p_{Tukey} = 0.367; d = 0.34; BF_{10} = 0.85)$. This pattern of group differences suggests

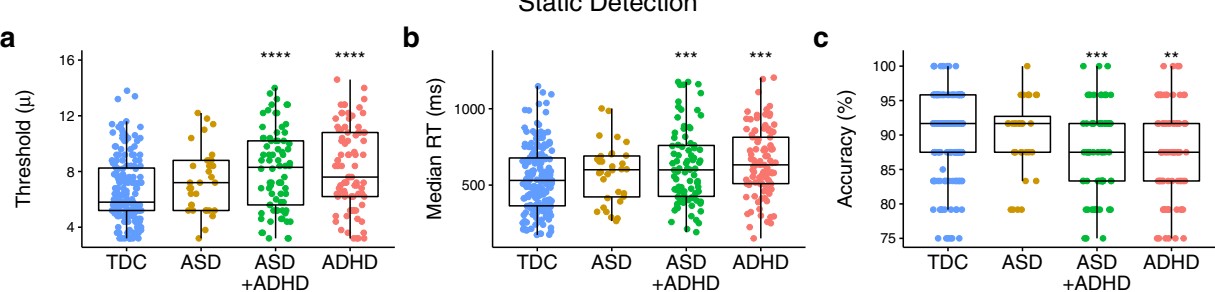

**Fig. 2 Group comparisons made on performance outcomes from the static detection protocol.** Individual data points with overlaid boxplots visualizing the group comparisons on key performance outcomes on the detection protocol, including **a** static detection threshold, **b** reaction time, and **c** accuracy. From these plots, it is clear that when compared to the typically developing control and ASD groups, static detection thresholds are significantly elevated in the ASD + ADHD and ADHD groups. Note that all statistical analyses presented in the main text of the manuscript for the plots presented in this figure are age-corrected. Values in the plots presented within this figure are not corrected for age due to the loss of interpretability of the units of measurement (which occurs when presenting residual rather than raw values). The precise number of participants in each group and further descriptive statistics can be found in Supplementary Methods, Supplementary Table 1. *$p < 0.05$, **$p < 0.01$, ***$p < 0.001$, TDC typically developing controls, ASD autism spectrum disorders, ADHD attention-deficit hyperactivity disorder, RT reaction times.

difficulties with detecting tactile stimuli are more common to children with ADHD. While children in the ADHD and ASD + ADHD had higher detection thresholds than the typically developing control group, those with a diagnosis of ASD without a co-diagnosis of ADHD, did not. This finding is generally consistent with earlier work conducted both in our lab[16,33,34] and others[34,35]. A point of difference in the current study, however, is that elevated detection thresholds were only identified in the ASD group with a co-diagnosis of ADHD.

### Discrimination

*Amplitude discrimination.* Results from group comparisons on the performance outcomes of the amplitude discrimination protocol are plotted in Fig. 3. There was a significant main effect of group on amplitude discrimination thresholds ($F(3, 395) = 5.79$, $p < 0.001$; $\eta^2_p = 0.04$; $BF_{10} = 14,610.86$) and a non-significant trend towards a main effect of group on accuracy ($F(3, 358) = 2.39$, $p = 0.069$; $\eta^2_p = 0.02$; $BF_{10} = 3.40$). There was no significant main effect of group on median reaction times ($F(3, 390) = 1.31$, $p = 0.269$; $\eta^2_p = 0.01$; $BF_{10} = 0.08$), or the number of reversals ($F(3, 366) = 1.90$, $p = 0.128$; $\eta^2_p = 0.02$; $BF_{10} = 0.03$). Post hoc comparisons found that while individuals in the ASD + ADHD ($t(395) = 4.02$, $p_{Tukey} < 0.001$; $d = 0.70$; $BF_{10} = 258.97$) group had significantly higher amplitude discrimination thresholds than those in the typically developing control group, those in the ASD ($t(395) = 1.41$, $p_{Tukey} = 0.490$; $d = 0.34$; $BF_{10} = 0.50$) and ADHD ($t(395) = 2.24$, $p_{Tukey} = 0.113$; $d = 0.60$; $BF_{10} = 1.62$) groups did not. The ASD + ADHD group was also less accurate than children in the typically developing control group ($t(358) = 2.65$, $p_{Tukey} = 0.041$; $d = 0.70$; $BF_{10} = 3.57$). The same pattern was observed when the amplitude discrimination protocols were completed with single and dual-site adaptation (see Supplementary Results, Supplementary Fig. 5).

*Frequency discrimination.* Results from group comparisons on the performance outcomes of the frequency discrimination protocol are shown in Fig. 4. There was a significant main effect of group on frequency discrimination thresholds ($F(3, 399) = 4.40$, $p = 0.005$; $\eta^2_p = 0.03$; $BF_{10} = 63.40$) and accuracy ($F(3, 399) = 3.57$ $p = 0.014$; $\eta^2_p = 0.03$; $BF_{10} = 20.85$). There was also a non-significant trend towards a main effect of group on median reaction times ($F(3, 377) = 2.13$, $p = 0.096$; $\eta^2_p = 0.02$; $BF_{10} = 0.36$). There was no significant main effect of group on the number of reversals ($F(3, 355) = 0.45$, $p = 0.721$; $\eta^2_p = 0.003$; $BF_{10} = 0.008$). Compared to the typically developing control group, frequency discrimination

thresholds were significantly worse in the ASD + ADHD ($t(430) = 3.60$, $p_{Tukey} = 0.002$; $d = 0.70$; $BF_{10} = 58.45$) group. The ASD ($t(399) = 1.16$, $p_{Tukey} = 0.654$; $d = 0.34$; $BF_{10} = 0.38$) and ADHD ($t(399) = 1.49$, $p_{Tukey} = 0.444$; $d = 0.59$; $BF_{10} = 0.43$) groups otherwise had similar frequency discrimination thresholds to those in the typically developing control group. Only the ASD + ADHD group were less accurate than the typically developing control group ($t(399) = 3.27$, $p_{Tukey} = 0.006$; $d = 0.70$; $BF_{10} = 20.12$). While not presented here, the same pattern of results was identified for group comparisons made on performance outcomes when the stimuli were delivered sequentially (see Supplementary Results, Supplementary Fig. 4).

When the results of the group comparisons from the discrimination protocols are taken together, difficulties with tactile discrimination appear to be more common to children with ASD + ADHD. Indeed, both amplitude and frequency discrimination thresholds were only significantly higher in the ASD + ADHD group, with no group differences being observed between the ASD and ADHD groups to the typically developing control group for either of the discrimination thresholds. While these results might suggest an additive effect of co-diagnosis, such that having both ASD and ADHD results in an elevation of discrimination thresholds not otherwise seen in ASD and ADHD, we note that the group comparisons between the ASD + ADHD to the ASD and ADHD groups did not identify any meaningful differences. Regardless of the interpretation, it is interesting to note that both amplitude and frequency discrimination thresholds are elevated in ASD + ADHD, suggesting an impairment of a shared neurophysiological process. Indeed, amplitude and frequency discrimination are correlated (see Supplementary Results, Supplementary Fig. 6), supporting the suggestion that they may rely on at least partially overlapping cortical processes.

*Order judgement.* We first note that a smaller sample (~64% of the total) completed the order judgement protocol, as this was introduced into the protocol later into data collection (see Supplementary Methods, Supplementary Table 1). Results from group comparisons on the performance outcomes of the temporal order judgement protocol are plotted in Fig. 5. After accounting for age, there was evidence for an effect of group on order judgement thresholds ($F(3, 260) = 2.55$, $p = 0.056$, $\eta^2_p = 0.03$; $BF_{10} = 1287.39$) and accuracy ($F(3, 247) = 1.87$, $p = 0.135$, $\eta^2_p = 0.06$; $BF_{10} = 1,034,282.53$). There were no group effects on median reaction times ($F(3, 262) = 1.42$, $p = 0.237$, $\eta^2_p = 0.01$; $BF_{10} = 0.01$) or the number of reversals ($F(3, 252) = 1.61$,

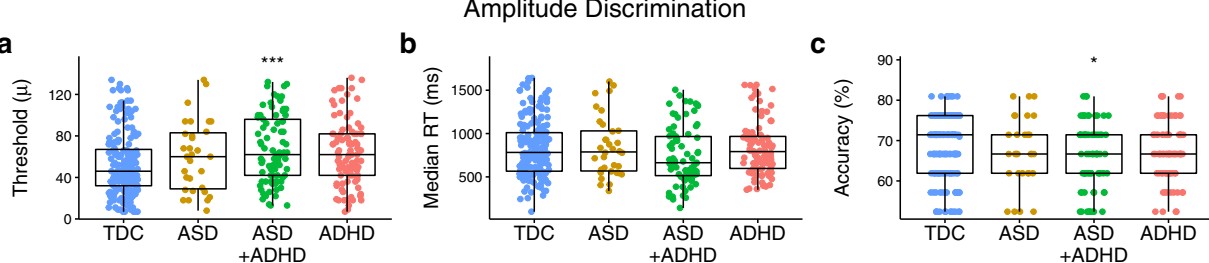

**Fig. 3 Group comparisons made on performance outcomes from the amplitude discrimination protocol.** Individual data points with overlaid boxplots visualizing the group comparisons on key performance outcomes on the amplitude discrimination protocol, including **a** amplitude discrimination threshold, **b** reaction time, and **c** accuracy. While it appears that all groups have elevated amplitude discrimination thresholds compared to the typically developing control group, after controlling for age, only the ASD + ADHD had significantly higher amplitude discrimination thresholds than the typically developing control group. Note that all statistical analyses presented in the main text of the manuscript for the plots presented in this figure are age-corrected. Values in the plots presented within this figure are not corrected for age due to the loss of interpretability of units of measurement that occurs when presenting residual rather than raw values. The precise number of participants in each group and further descriptive statistics can be found in Supplementary Methods, Supplementary Table 1. *$p < 0.05$, **$p < 0.01$, ***$p < 0.001$, TDC typically developing controls, ASD autism spectrum disorders, ADHD attention-deficit hyperactivity disorder, RT reaction times.

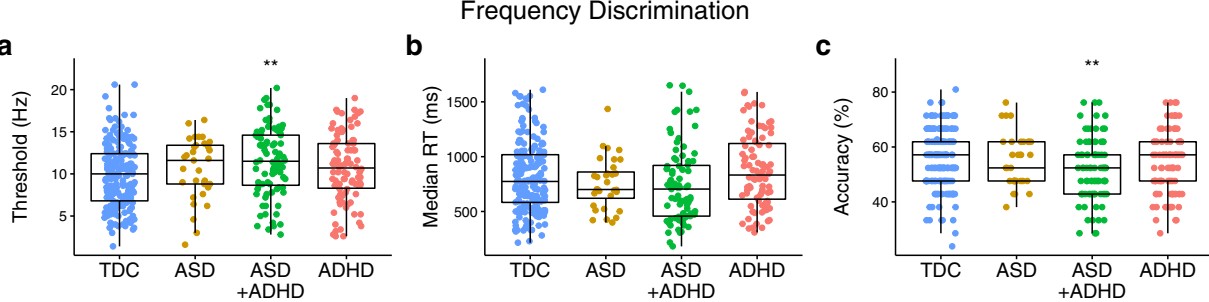

**Fig. 4 Group comparisons made on performance outcomes from the frequency discrimination protocol.** Individual data points with overlaid boxplots visualizing the group comparisons on key performance outcomes on the frequency discrimination protocol, including **a** frequency discrimination threshold, **b** reaction time, and **c** accuracy. As with amplitude discrimination thresholds, after controlling for age, only the ASD + ADHD group had higher frequency discrimination thresholds than the typically developing control group. Note that all statistical analyses presented in the main text of the manuscript for the plots presented in this figure are age-corrected. Values in the plots presented within this figure are not corrected for age due to the loss of interpretability of units of measurement that occurs when presenting residuals rather than raw values. The precise number of participants in each group and further descriptive statistics can be found in Supplementary Methods, Supplementary Table 1. *$p < 0.05$, **$p < 0.01$, ***$p < 0.001$, TDC typically developing controls, ASD autism spectrum disorders, ADHD attention-deficit hyperactivity disorder, RT reaction times.

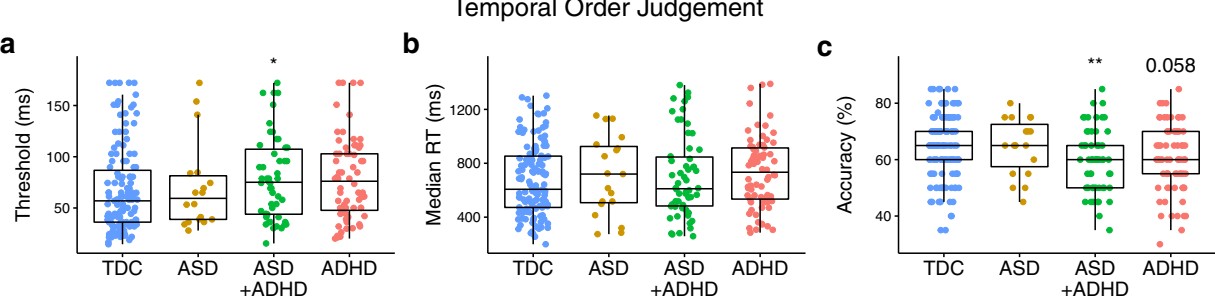

**Fig. 5 Group comparisons made on performance outcomes from the temporal order judgement protocol.** Individual data points with overlaid boxplots visualizing the group comparisons on key performance outcomes on the temporal order judgement protocol, including **a** order judgement threshold, **b** reaction time, and **c** accuracy. While it would appear that only the ASD + ADHD and ADHD groups had elevated order judgement thresholds compared to the typically developing control group, after controlling for age, only the ASD + ADHD group had higher order judgement thresholds compared to controls. However, when looking at accuracy, the ASD + ADHD and ADHD group showed significant and a non-significant trend towards being less accurate than the typically developing control group (respectively). Note that all statistical analyses presented in the main text of the manuscript for the plots presented in this figure are age-corrected. Values in the plots presented within this figure are not corrected for age due to the loss of interpretability of units of measurement that occurs when presenting residuals rather than raw values. The precise number of participants in each group and further descriptive statistics can be found in Supplementary Methods, Supplementary Table 1. *$p < 0.05$, **$p < 0.01$, ***$p < 0.001$.

$p = 0.186$, $\eta^2_p = 0.01$; $BF_{10} = 0.07$). Order judgement thresholds were significantly higher (i.e., a wider temporal binding window) in ASD + ADHD group compared to those in the typically developing control group ($t(260) = 2.65$, $p_{Tukey} = 0.042$; $d = 0.48$; $BF_{10} = 4.48$). Order judgement thresholds in the ASD ($t(260) = 0.77$, $p_{Tukey} = 0.869$; $d = 0.25$; $BF_{10} = 0.33$) and ADHD groups ($t(260) = 1.51$, $p = 0.433$; $d = 0.24$; $BF_{10} = 0.47$) were otherwise comparable to those in the typically developing control group. Compared to the typically developing control group, accuracy was significantly lower in the ASD + ADHD group ($t(288) = 3.78$, $p_{Tukey} = 0.001$; $d = 0.60$; $BF_{10} = 124.36$) and there was some evidence the ADHD group was also less accurate than controls ($t(288) = 2.53$, $p_{Tukey} = 0.058$; $d = 0.21$; $BF_{10} = 2.66$).

The pattern of results here suggests that difficulties with order judgement are more common to individuals with ADHD, with the ASD + ADHD group having higher order judgement thresholds than the typically developing control group, and with both the ASD + ADHD and ADHD groups also being less accurate. Higher order judgement thresholds or "wider temporal binding windows", in which two temporally separate stimuli are perceived as one[36], has also been identified in ADHD across other sensory domains, including visual and auditory domains[36–38], suggesting a general deficit of time perception[39], rather than a tactile specific alteration. Further, as stated earlier, wider temporal binding windows have also been identified in ASD[15–19], though the findings were mixed[17,20–26]. Given that elevated order judgement thresholds were only identified in the ASD + ADHD and ADHD groups, it is possible that discrepancies amongst earlier studies could be related to a lack of consideration of co-occurring ADHD symptomatology.

**Associations between tactile sensitivity and sensory processing and experience.** Beyond identifying differences in tactile sensitivity between groups, we were interested in determining whether individual differences in each of these processing domains were related to caregiver reported difficulties with a child's processing and experience of sensory stimuli, which have historically been more commonly used to characterize sensory abnormalities in ASD and ADHD. Here we conducted multiple correlation analyses between tactile thresholds and subscale items on the on the Sensory Processing Measure[40] and Sensory Experience Questionnaire[41]. After adjusting for multiple comparisons using the Bonferroni method (see "Methods" section for full details), detection thresholds showed the strongest and most significant associations to items of the Sensory Processing Measure and Sensory Experience Questionnaire (Fig. 6).

Individuals with higher detection thresholds had more caregiver reports of problems with social and non-social combined scores of sensory hyper-responsivity ($r = 0.28$, $p_{Bonferroni} = 0.005$) and hypo-responsivity ($r = 0.37$, $p_{Bonferroni} < 0.0001$) and were reported to display more sensory seeking behaviors ($r = 0.31$, $p_{Bonferroni} = 0.003$). Children with higher detection thresholds also had more reported problems with social participation ($r = 0.22$, $p_{Bonferroni} = 0.006$) and body awareness ($r = 0.29$, $p_{Bonferroni} < 0.001$). These associations provide support to the common suggestion that alterations in "lower-order" fundamental aspects of sensory processing could explain some of the more complex "higher-order" behavioral social problems experienced by those with ASD and ADHD[27,42,43].

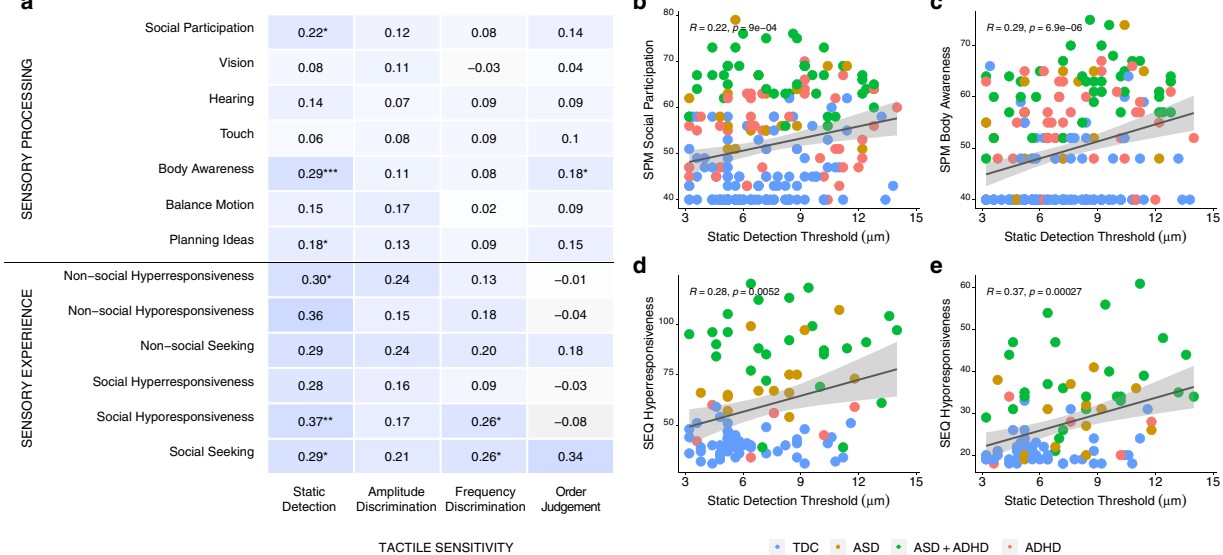

**Fig. 6 Correlation analyses between tactile sensitivity thresholds and subscale items from the Sensory Processing Measure and Sensory Experience Questionnaire. a** Heatmap correlations between subscales of the Sensory Processing Measure and Sensory Experience Questionnaire (y-axis) and relevant tactile sensitivity metrics across each subdomain (x-axis). The "*" are used to denote statistical significance after correcting for multiple correlations using Bonferroni's correction (*$p < 0.05$, **$p < 0.01$, ***$p < 0.001$). Due to the large number of correlations being conducted, Bonferroni corrections were applied to each of the correlation analyses conducted by multiplying the resulting $p$-value of each correlation between a given tactile sensitivity metric by the number of subscale items within each questionnaire. For example, for all of the correlations conducted between the Sensory Processing Measure and static detection thresholds, the resulting $p$-value was multiplied by 7 (i.e., number of subscale items within the Sensory Processing Measure). It is clear that the tactile sensitivity metric that was most often associated with the subscale items of the Sensory Processing Measure and Sensory Experience Questionnaire questionnaires was static detection thresholds. On the right we highlight the correlations between static detection thresholds and **b** social participation, **c** body awareness, **d** hyper-responsiveness and **e** hypo-responsiveness. Note that the scatterplots presented in **d** and **e** are using combined scores of social and non-social hyper-responsiveness and hypo-responsiveness (which are otherwise presented individually on the heatmap in **a**). See Supplementary Results, Supplementary Fig. 7 for additional correlation analyses. Note that the $p$-values (*$p < 0.05$, **$p < 0.01$, ***$p < 0.001$) presented within each scatterplot are the unadjusted $p$-values. TDC typically developing controls, ASD autism spectrum disorders, ADHD attention-deficit hyperactivity disorder, RT reaction times, SPM sensory processing measure, SEQ sensory experience questionnaire.

**Associations between tactile sensitivity and disorder-specific clinical symptomatology.** Next, to determine whether individual differences in tactile sensitivity were related to disorder-specific clinical symptomatology, we conducted multiple correlation analyses between the tactile thresholds and subscale scores of clinical questionnaires specific to each disorder, including the Autism Diagnostic Interview-Revised[44] and Autism Diagnostic Observation Scale[45] for ASD-specific symptomatology and the Conners[46], Conners 3rd edition[47], and DuPaul's ADHD Parent Rating Scale for ADHD-specific symptomatology.

*Tactile sensitivity and ASD-specific symptomatology.* We wish to first note that since only the children who had a primary diagnosis of ASD (i.e., those in the ASD and ASD + ADHD groups) were administered the Autism Diagnostic Observation Scale and had parents/caretakers complete the Autism Diagnostic Interview-Revised, children with a primary diagnosis of ADHD are not included these analyses. After correcting for multiple comparisons, only two associations between the tactile sensitivity thresholds and items from the ASD-specific subscales remained significant (see Fig. 7a). Namely, both amplitude and frequency discrimination thresholds were positively correlated to

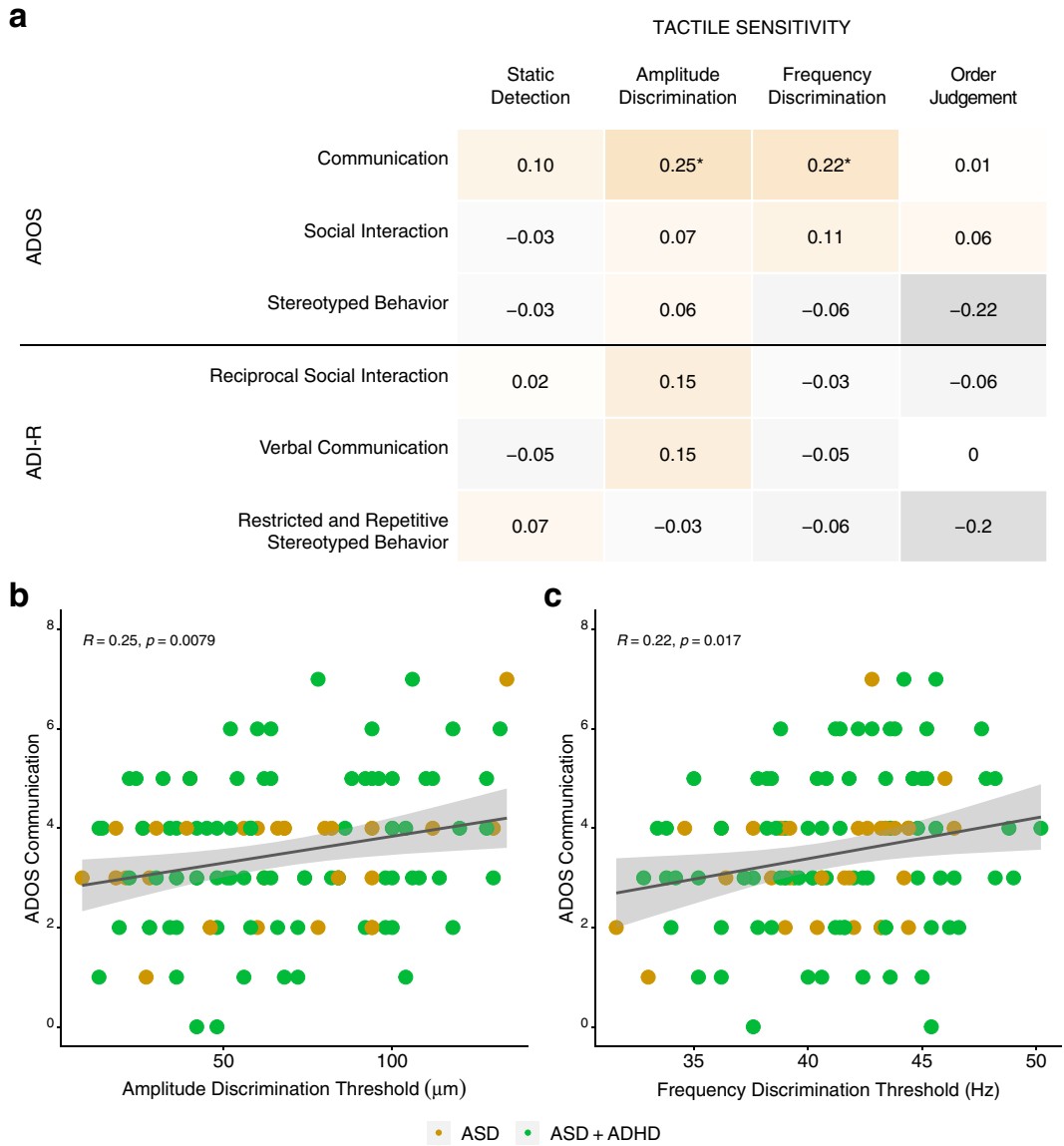

**Fig. 7 Correlation analyses between tactile sensitivity thresholds and total domain scores from the Autism Diagnostic Observation Schedule and Autism Diagnostic Interview-Revised. a** Heatmap correlations between subscales of the Autism Diagnostic Observation Scale and Autism Diagnostic Interview-Revised (*y*-axis) and relevant tactile sensitivity metrics (*x*-axis). The "*" are used to denote statistical significance after correcting for multiple correlations using Bonferroni's correction (*$p < 0.05$, **$p < 0.01$, ***$p < 0.001$). Due to the large number of correlations being conducted, Bonferroni corrections were applied to each of the correlation analyses conducted by multiplying the resulting *p*-value of each correlation between a given tactile sensitivity metric by the number of subscale items within each questionnaire. Two associations survived Bonferroni corrections and are presented in the bottom row of the figure. There were positive correlations between problems with communication on the Autism Diagnostic Observation Scale with (**b**) amplitude discrimination thresholds and (**c**) frequency discrimination thresholds. See Supplementary Results, Supplementary Fig. 8 for additional correlation analyses. Note that the *p*-values (*$p < 0.05$, **$p < 0.01$, ***$p < 0.001$) presented within each scatterplot are the unadjusted *p*-values. ASD autism spectrum disorders, ADHD attention-deficit hyperactivity disorder, ADOS Autism Diagnostic Observation Scale, ADI-R Autism Diagnostic Interview-Revised.

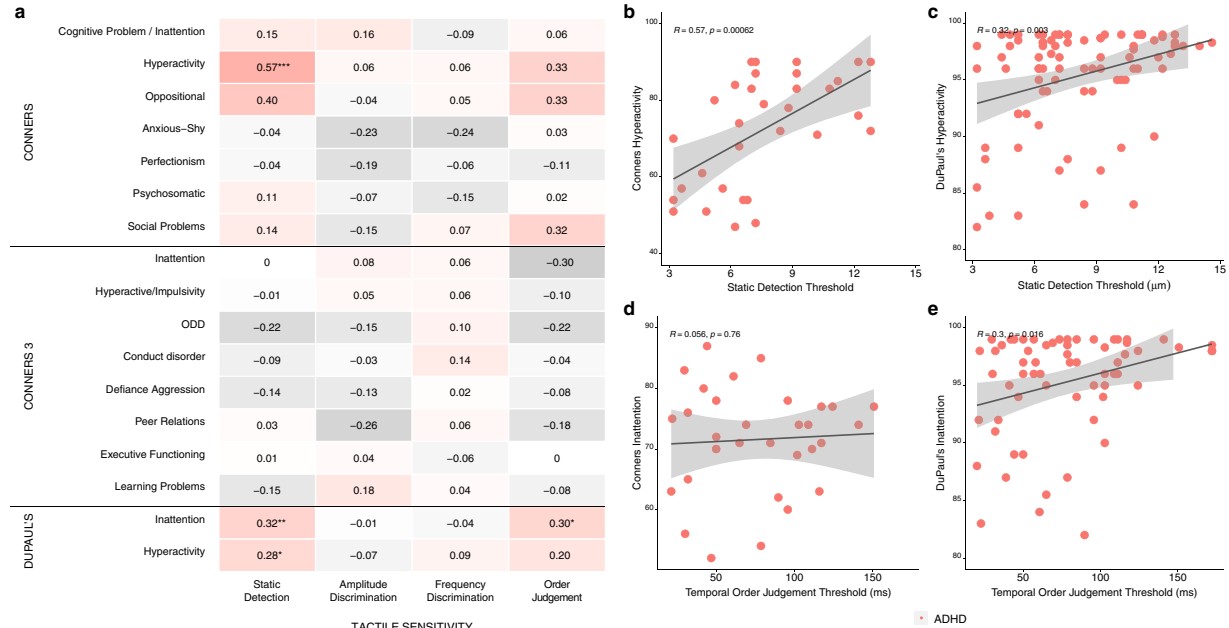

**Fig. 8 Correlation analyses between tactile sensitivity thresholds and items from the Conners, Conners 3rd Edition and DuPaul's ADHD rating scales.**
**a** Heatmap correlations between subscales of the Conners, Conners 3, and DuPaul's ADHD rating scales (y-axis) and relevant tactile sensitivity metrics (x-axis). The "*" are used to denote statistical significance after correcting for multiple correlations using Bonferroni's correction (*$p < 0.05$, **$p < 0.01$, ***$p < 0.001$). "t" was used to denote a non-significant trend (i.e., $p < 0.10$). Due to the large number of correlations being conducted, Bonferroni corrections were applied to each of the correlation analyses conducted by multiplying the resulting p-value of each correlation between a given tactile sensitivity metric by the number of subscale items within each questionnaire. After corrections were applied, there were six significant and two non-significant trending correlations. We have selectively chosen to highlight the most meaningful correlations as scatterplots presented on the right side. Hyperactivity on the **b** Conner's and **c** DuPaul's rating scales correlated significantly with detection thresholds. Inattention on the **d** Conner's rating scale was not significantly correlated to order judgement thresholds, while **e** DuPaul's rating scales were. See Supplementary Results, Supplementary Fig. 9 for additional correlation analyses. Note that the p-values (*$p < 0.05$, **$p < 0.01$, ***$p < 0.001$) presented within each scatterplot are the unadjusted p-values. ADHD attention-deficit hyperactivity disorder, ODD oppositional defiant disorder.

communication subscale scores on the Autism Diagnostic Observation Scale, such that children with higher amplitude ($r = 0.25$, $p_{Bonferroni} = 0.023$), and frequency ($r = 0.22$, $p_{Bonferroni} = 0.050$) discrimination thresholds had more reported problems with communication on the Autism Diagnostic Observation Scale (see Fig. 7b, c, respectively).

*Associations between tactile sensitivity and ADHD symptomatology.* For the correlations between the tactile sensitivity thresholds and the items from the ADHD-specific questionnaires, only children who had a primary diagnosis of ADHD were included (i.e., those in the ASD + ADHD group were not included in these analyses). After correcting for multiple comparisons, significant associations between detection thresholds and hyperactivity subscales on both the Conners ($r = 0.57$, $p_{Bonferroni} < 0.010$) and DuPaul's rating ($r = 0.32$, $p_{Bonferroni} = 0.020$) scales remained (see Fig. 8b, c). A trend for an association between order judgement thresholds and inattention on the DuPaul's rating scale ($r = 0.30$, $p_{Bonferroni} = 0.030$) also remained (see Fig. 8e). Unlike detection thresholds and hyperactivity, there was no cross-questionnaire congruence, with inattention on the Conners rating scale not being meaningfully or significantly related to order judgement thresholds ($r = 0.06$, $p_{Bonferroni} = 0.999$). See Fig. 8d.

When taken together, our results suggest that children with ASD and ADHD have distinct patterns of altered tactile perception. These perceptual alterations also appear to be exclusively related to disorder-specific symptom severity. For example, detection thresholds, which was more commonly elevated in those with either a primary or secondary diagnosis of ADHD, positively scaled with hyperactivity. Indeed, the children that had higher detection thresholds were also the children that had more severe hyperactivity scores on the ADHD-specific clinical scales (i.e., the Conners and DuPaul's ADHD rating scale). Importantly, although detection thresholds were associated with items from the Sensory Processing Measure and Sensory Experience Questionnaire, detection thresholds were not meaningfully correlated to any of the items on the ASD-specific clinical scales (i.e., the Autism Diagnostic Observation Scale). Following the pattern with the ADHD-specific alterations, amplitude and frequency discrimination thresholds, which had shown a pattern of being more commonly elevated in those with ASD, scaled exclusively with communication scores on the Autism Diagnostic Observation Scale, such that children who had higher discrimination thresholds were also those who had more reported problems with communication. This general pattern was also found for order judgement thresholds, such that elevated order judgment thresholds, more common in children with a diagnosis of ADHD, exclusively correlated with inattention scores on ADHD-specific clinical scales. Indeed, children with ADHD who had wider temporal binding windows (i.e, higher order judgement thresholds) also had more reported difficulties with inattention.

## Discussion
Prior to discussing the neurophysiological implications of our findings, it is worth reiterating that although we have identified disorder-specific patterns of alterations in the tactile domain, prior evidence from clinical cohorts[48,49] suggests that difficulties with processing in one sensory domain are usually accompanied

by similar difficulties processing in another. That is, although different senses are primarily processed by distinct sensory-specific areas of the brain[50] (e.g., tactile stimuli are initially processed in S1), certain aspects of sensory processing rely on shared brain structures and/or neuronal mechanisms[51,52]. In support of this, despite our finding of elevated tactile discrimination thresholds in ASD, atypical (either reduced or enhanced) discrimination of visual[53–55] and auditory[56,57] stimuli has also been identified in ASD. Similarly, the difficulties with order judgement that we identified in our ADHD and ASD + ADHD groups have also been identified when using visual and auditory stimuli[36,58]. In short, the disorder-specific alterations of tactile processing we identified likely reflect a dysfunction of more fundamental and generalized neurophysiological functions required for processing across sensory domains (i.e., they are domain-general), rather than a dysfunction of sensory processing specific to the tactile domain (i.e., domain-specific alterations). With that in mind, although the findings of the current study are restricted to tactile processing, it is possible (and perhaps likely based on prior evidence) that our findings would also hold across other sensory domains.

## Neurophysiological explanations of disorder-specific alterations of sensory perception

*Amplitude and frequency discrimination in ASD.* While many hypotheses have been proposed to explain the sensory alterations in ASD, the current prevailing hypothesis is that of abnormal GABA mediated neurotransmission[59–61]. With specific regard to difficulties with amplitude and frequency discrimination in the current study, postmortem comparisons of the brains of adults with and without ASD have found that that the brains of adults with ASD had fewer lateral inhibitory connections between cortical minicolumns[62]. Given that lateral inhibition via GABAergic interneurons between cortical minicolumns is fundamental for appropriate discrimination of sensory input[63], our finding of specific difficulties with tactile discrimination in children with ASD is consistent with the broad idea of atypical GABA inhibition in ASD. We have also previously identified an association between tactile discrimination thresholds and local GABA levels in S1 in children with ASD[34].

It is important not to understate the importance of GABA for sensory perception as a whole. Just as GABAergic lateral inhibition is required for sharpening the receptive fields of tactile sensory input to improve discrimination[64], GABA is also required for discrimination in other sensory modalities. For example, in the visual system, lateral inhibition is used to facilitate edge detection, which in turn improves the contrast and resolution of visual stimuli[65]. Similarly, given that tonotopic organization is maintained throughout the auditory system[66,67], lateral inhibition of neighboring cells that respond to similar frequencies to those being actively processed is required for accurate tone discrimination. Beyond lateral inhibition, fast-spiking $GABA_A$ receptors (which are suggested to be affected in ASD[68]) also play a critical role in synchronizing spike timing of cortical neurons involved in encoding spatiotemporal characteristics of sensory stimuli[69]. Given the many roles that GABA plays in sensory perception, it should not be surprising that dysfunction of the GABA system could have wide ranging implications for overall neurodevelopment.

While speculative, it is possible that dysfunction of the GABA system, which in turn results in alterations of lower-order and domain-general alterations of sensory processing, contributes to many of the higher-order behavioral symptoms of ASD. Take for instance, the deficits in social communication and interaction that is characteristic of ASD. Although regions such as the medial prefrontal cortex and superior temporal sulcus are more typically referred to when discussing the "social brain"[70], social cognition is dependent on downstream input from sensory regions. If the sensory information is not appropriately processed (and hence compromised), then the capacity of the medial prefrontal cortex and superior temporal sulcus to make use of that information may be limited. Indeed, many important social cues require a degree of sensory discrimination. For example, emotions in speech are conveyed by slight changes in both the amplitude and frequency of tone[71]. One can imagine that difficulties with discriminating in the auditory domain could then result in failures to identify the correct emotion behind spoken sentences. Similarly, in the visual domain, difficulties with visual discrimination could also conceivably affect accurate identification of emotions communicated through small changes in facial expressions. Indeed, atypical tone[57,72,73] and face[74–76] discrimination have been previously identified in ASD. Especially from the perspective of developmental stages, altered sensory processing in critical periods of development (e.g., infancy and adolescence) could have serious long-term consequences on overall social functioning.

*Detection and order judgement in ADHD and ASD + ADHD.* Elevated detection thresholds have been associated with lower levels of S1 GABA in ASD[77] (though like many previous studies, co-occurring ADHD was not considered). It is difficult to speculate why GABA may associate with either differences in detection or discrimination, since GABA is involved in both. However, magnetic resonance spectroscopy of GABA is only one measure pertaining to the GABAergic mechanism whereas differences in detection or discrimination could be indicative of alterations in a number of processes related to GABAergic inhibition. While associations between elevated detection thresholds and S1 GABA have not yet been identified in ADHD, magnetic resonance spectroscopy studies have shown atypical GABA in multiple brain regions in those with ADHD[78–80]. Similarly, evidence from transcranial magnetic stimulation studies have also suggested atypical GABAergic receptor functioning of the cortex in children with ADHD[81,82]. In our own work, we have also previously identified an association between elevated detection thresholds and reduced S1 GABA levels in Tourette's syndrome[77,83], a disorder showing high comorbidity rates with ADHD[84]. As a whole, these data indicate the possibility that S1 GABA, in either its tonic availability or its phasic functioning at the receptor level, is involved in ADHD pathophysiology and may explain the elevated detection thresholds seen in the current study[85,86].

With respect to elevated order judgement thresholds, timing deficits are a consistent finding in individuals with ADHD and appear to be present in their performance across a wide range of behavioral paradigms associated with the perception of time, but not specifically related to order judgement[87,88]. For example, even within the findings from the group comparisons made on the reaction time protocols from our battery (see Supplementary Results), children with ADHD and ASD + ADHD showed significantly longer and more variable reaction times than typically developing controls when required to make a speeded response to suprathreshold tactile stimuli. Dopamine is commonly implicated in timing functions[89,90] and in the neurobiology of ADHD[91]. Treatment using dopaminergic psychostimulants in ADHD are also found to improve both order judgement thresholds and reduce reaction time variability[90], suggesting that elevated order judgement thresholds arise out of dysfunction of the dopaminergic system in those with ADHD.

## Hyper-responsivity and hypo-responsivity, a false dichotomy?
The associations between static detection thresholds and both

hyper-responsivity and hypo-responsivity are of strong interest. First, they broadly suggest that tactile sensitivity assessed using psychophysics correspond with caregiver reports of problems with sensory processing, showing concurrent validity. Second, detection thresholds being associated with both hypersensitivity and hyposensitivity is interesting in that this would appear to run counter to the common narratives of hyper-responsivity and hypo-responsivity as either distinct traits or opposite extremes along a single axis. In fact, we found that hyper-responsivity and hypo-responsivity and sensory seeking were all strongly and positively correlated ($r > 0.75$, $p_{\text{Bonferroni}} < 0.001$ for all three combination of correlations, see Supplementary Results, Supplementary Fig. 11). It is therefore important to emphasize that children who are "hypersensitive" often score high on hyposensitivity scales (as well as sensory seeking scales). We suggest that we should no longer differentiate between hyper-responsivity and hypo-responsivity, as 1) notions of the hyper-responsive or hypo-responsive phenotypes appear untrue, and 2) our data suggest that abnormalities in tactile perception (i.e., elevated detection thresholds) link to both hyper-responsive and hyporesponsive behaviors. Note that it is possible that hypo-responsivity and hyper-responsivity to sensory stimuli has a temporal component, where factors such as time of day and duration of exposure has effects on whether individuals are hypo- or hyperresponsive to a given stimulus. Additionally, given that sensory alterations are unlikely to be domain-specific (as we have discussed above above), it is likely that hypo-responsivity and hyper-responsivity to stimuli is also domain-general, though this has not yet been thoroughly investigated.

**Limitations and future directions**. This work is not without limitations. For instance, while we have identified group differences, it is clear that there is a notable amount of group overlap in the outcome measures (see Supplementary Discussion for further elaboration). Moreover, while we assessed a range of protocols to assess different aspects of tactile sensitivity, the protocols adopted here are not exhaustive. For instance, while all of the tactile stimuli delivered in our battery were in the flutter range (0–50 Hz), it is unclear whether the pattern of results identified here would hold for higher frequency stimuli. Indeed, different cortical mechanisms may underlie even very similar tasks with slightly different parameters (e.g., amplitude discrimination of high frequency stimuli may require different cortical processes than amplitude discrimination of lower frequency stimuli). Future investigations taking a more mechanistic approach to understanding how and why these alterations manifest at the neural level (e.g., to what extent, or why, is discrimination different in ASD; or what role does excitation/inhibition balance play in state-dependent processing of stimuli) have the potential to inform treatment development at an individual, rather than at a diagnostic level.

To summarize, the results of this study point to disorder-specific alterations of tactile processing in ASD and ADHD, showing that tactile perceptual differences are uniquely related to the core symptoms of each disorder. Children with ADHD that had higher detection and order judgement thresholds also had greater problems with hyperactivity and inattention. Similarly, children with ASD that had higher discrimination thresholds also had greater problems with social communication. When taken together, these findings highlight the relevance of sensory, specifically tactile, processing alterations to the core symptoms of ASD and ADHD, supporting theories that altered sensory processing may cause or exacerbate core symptoms of these disorders. The relationships between these more fundamental aspects of tactile processing (i.e., perceptual thresholds) and

symptom expression in these disorders would suggest tactile processing alterations to be useful biomarkers of disorder-specific symptom severity. While sensory alterations have been described previously, here we show distinct and shared sensory alterations between ASD and ADHD that may inform future approaches in how we diagnose and treat these different disorders.

## Methods

**Participants**. A total of 434 children were included in this study, of which 197 were typically developing controls, 34 had ASD (but no diagnosis of ADHD), 99 had both ASD + ADHD, and 104 had ADHD (but no diagnosis of ASD). We note that while the frequency of comorbidity (i.e., ASD + ADHD occurred in ~74% of children who met criteria for ASD) in our sample may seem uncharacteristic of the true frequency of comorbidity in the population, a consensus estimate of the true frequency of comorbidity in the population has not been established. Studies have reported comorbidity rates between 40 and 76%[92–94]. A summary of the relevant descriptive variables, including age, sex, and handedness for each group that met screening/inclusion/exclusion criteria (described in detail below) are presented in Table 1. Informed consent was obtained from a caregiver of each assenting child that was tested. Studies in which these data were collected were/are under the ethical approval of the Kennedy Krieger Institute and the Johns Hopkins School of Medicine Institutional Review Boards.

*Recruitment, screening, and general exclusion criteria*. Participants were recruited through local schools, with additional resources including community-wide advertisements, advocacy organizations, and medical institutions. Screening was performed via telephone interview with the parent or caregiver of each child. Children with a history of intellectual disability, seizures, brain injury, or other neurological disorder (i.e., Tourette's syndrome) or illnesses were excluded. Children who were taking psychotropic medications other than stimulants were excluded from participation. For children who were on stimulant medication, children temporarily ceased taking stimulant medication on the day prior to and the day of testing. Handedness was evaluated using the Edinburgh Handedness Inventory[95]. Diagnostic status was confirmed or established using the criteria described below.

### Diagnostic criteria
*Typically developing controls*. All typically developing controls were free of criteria for psychiatric disorders of the Diagnostic Interview for Children and Adolescents Fourth Edition (DICA-IV)[96] and/or the Kiddie Schedule for Affective Disorders and Schizophrenia[97].

*Diagnosis of ASD*. Children in the ASD cohort met the Diagnostic and Statistical Manual of Mental Disorders Fourth (DSM-IV)[98] and/or Fifth Edition (DSM-V) criteria for ASD. This was confirmed with the Autism Diagnostic Observation Schedule-Generic[99] or Autism Diagnostic Observation Schedule Second Edition[45] and the Autism Diagnostic Interview-Revised[44]. The information was reviewed and verified by S. H. M., a child neurologist with over two decades of experience in diagnosing ASD in clinic and research settings. Any children in the ASD cohort with identifiable causes of autism (e.g., Fragile X syndrome) were excluded.

Intellectual ability was assessed with the Wechsler Intelligence Scale of Children-Fourth[100] or Fifth[101] edition. Children with full-scale IQ scores below 80 were excluded from participation unless there was a 12-point or greater index discrepancy, in which case either the Verbal Comprehension Index or Perceptual Reasoning Index (or Fluid Reasoning Index and Visual Spatial Index if the child was assessed using the Wechsler Intelligence Scale of Children Fifth edition) was required to be over 80 and the lower of the two was required to be over 65. To rule out a learning disability in reading, children performed the basic reading subtest from the Wechsler Individual Achievement Test[102] or the Wechsler Individual Achievement Test 2[103]. Those who demonstrated a significant discrepancy between full-scale IQ and the Wechsler Individual Achievement Test[102] or Wechsler Individual Achievement Test 2 score or a basic reading subtest below 85 were excluded from the study. To rule out children who met criteria for psychiatric disorders other than ASD and ADHD, children were required to complete the Diagnostic Interview for Children and Adolescents Fourth Edition and/or Kiddie Schedule for Affective Disorders and Schizophrenia. If children met criteria for conduct disorder, mood disorders, generalized anxiety disorder, separation anxiety disorder, social phobia, or obsessive-compulsive disorder, they were excluded from the study.

*Diagnosis of ADHD*. Children in the ADHD group met the DSM-IV and/or DSM-V criteria for ADHD. Specifically, children in the ADHD group must have had: 1) a T score of 60 or higher on scale L (DSM-IV: inattentive) or M (DSM-IV: hyperactive-impulsive) on the Conners or Conners 3[47] when available, or a score of 2 or 3 on at least 6 out of 9 items on the inattentive or hyperactivity/impulsivity scales of the ADHD-Rating Scale-IV; and 2) an ADHD diagnosis on the Diagnostic Interview for Children and Adolescents Fourth Edition and/or Kiddie Schedule for

Affective Disorders and Schizophrenia. Further information was obtained through the Conners or Conners 3 Parent and Teacher Rating Scales-Revised: Long Form [ADHD-specific broad behavior rating scales and the ADHD Rating Scale-IV, home and school versions (ADHD-RS or DuPaul scale)]. The information was reviewed and then diagnosis was verified by S. H. M., a child neurologist with over two decades of experience in diagnosing ADHD in clinic and research settings. Children with a comorbid diagnosis of oppositional defiance disorder (ODD; based on the Diagnostic Interview for Children and Adolescents—Fourth Edition and/or Kiddie Schedule for Affective Disorders and Schizophrenia) were not excluded, since ODD does not represent a separate subtype of ADHD.

*Diagnosis of comorbid ASD and ADHD.* Children who met criteria for both ASD and ADHD were considered to have comorbid ASD and ADHD.

*Psychophysical assessment of vibrotactile perception.* Tactile perception was assessed via two-alternative forced choice paradigms delivered using a Cortical Metrics four-digit tactile stimulator[104] (CM4). The CM4 stimulator has four 5-mm cylindrical probes capable of delivering vibrotactile stimuli in the form of sinusoidal pulses. All stimuli were delivered between 0–350 μm and between 0–50 Hz. In the case of the vibrotactile assessments used in the current study, participants placed all fingers of their left hand over each of the probes (see Fig. 1a). For all protocols, vibrotactile stimuli were delivered to either left digit 2 or left digit 3 in the flutter range (0–50 Hz). Responses to each protocol were made using the homologous fingers of the contralateral side (i.e., right digit 2 and 3). A computer running custom Cortical Metrics scripts was used to visualize each trial and control the parameters of the vibrotactile stimuli (i.e., amplitude and frequency) of each trial for each protocol. Data were collected and stored on the same computer. The data were then later processed and visualized using a custom R package (available at: https://github.com/HeJasonL/BATD).

The full vibrotactile battery consisted of 11 protocols. For focus and clarity, in the main body of this manuscript we only report on static detection, amplitude and frequency discrimination, and temporal order judgment, as shown in Fig. 1. Additional protocols[105,106] that were completed by the participants, but are not reported in the main body of the manuscript are described in the Supplementary Methods. The results from the analysis of these data are also available as Supplementary Results. All the data and the code used to analyze these data are available on the OSF (https://osf.io/9mdc6/.).

To ensure participants had understood the protocol, each protocol condition was preceded by three consecutive practice trials that required correct responses to proceed. For all protocols, a stepwise adaptive tracking method was used (described in detail below). Participants could take as many breaks as possible except between protocols within the same domain.

*Static detection protocol.* A suprathreshold stimulus (frequency = 25 Hz; starting amplitude = 20 μm; duration = 500 ms) was pseudorandomly delivered to either left digit two or left digit 3. Participants were asked to respond on which finger they felt the stimulus. A one-up–one-down tracking paradigm (stimulus amplitude was decreased for a correct answer and increased for an incorrect answer) was used for the first ten trials and a two-up–one-down (two correct answers were necessary for a reduction in test amplitude) was used for the remainder of the task. There were 24 trials. Static detection threshold was calculated as the mean of the amplitudes of the last five trials.

*Amplitude discrimination protocol.* Stimuli were delivered to left digit 2 and left digit 3 simultaneously. One finger always received the standard stimulus (amplitude = 100 μm) while the other received the comparison stimulus (initial amplitude = 200 μm). The two stimuli were delivered to either digit pseudorandomly. Participants were asked to choose which of two simultaneously delivered stimuli had the higher amplitude (frequency = 25 Hz; duration = 500 ms; standard stimulus amplitude = 100 μm; initial comparison stimulus amplitude = 200 μm).

A one-up–one-down tracking paradigm (comparison stimulus amplitude was decreased by 10 μm for a correct answer and increased by 10 μm for a wrong answer) was used for the first ten trials and a two-up–one-down was used for the remainder of the task (20 trials delivered with a 5-s inter-trial interval). Amplitude discrimination thresholds were calculated as the mean of the amplitudes of the last five trials.

*Frequency discrimination protocol.* Stimuli were delivered to left digit 2 and left digit 3 simultaneously (duration = 500 ms; amplitude = 200 μm). One finger always received the standard stimulus (frequency = 30 Hz) while the other received the comparison stimulus (initial frequency = 40 Hz). The two stimuli were delivered to either digit pseudorandomly. Participants were asked which finger received the higher frequency stimulus.

A one-up–one-down tracking paradigm (the comparison stimulus frequency was decreased for a correct answer and increased for a wrong answer) was used for the first ten trials. A two-up–one-down was used for the remainder of the trials. All conditions contained 20 trials and were delivered with an ITI of 5 s. Frequency discrimination thresholds were obtained as the mean of the frequencies of the last five trials.

*Temporal order judgement protocol.* Each participant's ability to judge the order of sequentially delivered tactile stimuli was tested. Two single-cycle vibrotactile pulses (duration = 40 ms; frequency = 25 Hz; amplitude = 200 μm) were delivered to left digit 2 and left digit 3 separated temporally by a starting intertrial interval of 150 ms (the first pulse was assigned to either digit pseudorandomly) within a 1-s interval. A one-up–one-down tracking paradigm (the time between the sequentially delivered pulses was decreased by 10% for a correct answer and decreased by 10% for an incorrect answer). Participants were asked to make a response based on which digit they believed received the first pulse. The protocol contained 20 trials delivered with an ITI of 5 s. Temporal order judgment thresholds were calculated as the mean of the intertrial interval of the last five trials.

*Questionnaire assessment of sensory processing and experience.* Sensory processing and experience was assessed by administering the Sensory Processing Measure[40] and Sensory Experience Questionnaire[41] respectively. The Sensory Processing Measure is a norm-referenced assessment that produces scores for praxis, social participation, and sensory functioning across different sensory domains. The Sensory Experience Questionnaire was used in addition to the Sensory Processing Measure, since it was designed to evaluate the full range of sensory processing problems that are specific to ASD and provides an assessment of these problems in both social and nonsocial contexts. Both the Sensory Processing Measure and Sensory Experience Questionnaire have shown excellent reliability[107,108] and validity[41,109]. The Sensory Processing Measure has also shown convergent validity with another popular measure of sensory processing, namely the Sensory Profile[110].

**Statistics and reproducibility.** All statistical analyses were performed with R[111] (version 3.5.3). To determine whether the dependent variables of interest changed as a function of protocol condition within each group (i.e., a protocol effect), the dependent variables of interest for each condition pair were first compared within groups using repeated-measures linear mixed-effects models using the lme4 R package[112]. In all models, age and condition were included as fixed factors and participant was included as a random factor. Age was included as a covariate due to our preliminary analyses finding evidence for group differences in age (i.e., the ASD + ADHD group were slightly older than the ADHD group, $p = 0.011_{Bonferroni}$, the other groups were otherwise comparable in age $p > 0.358_{Bonferroni}$). Note that, while group differences were identified for IQ (as expected), IQ was not included as a covariate due to the reasons for not including IQ as a covariate in group comparisons for neurodevelopmental disorders described by Dennis and colleagues[113]. To test for significant effects, likelihood ratio tests were performed by comparing the log-likelihood statistic of the final models (i.e., the model including the dependent variable of interest, age, and condition as fixed factors and participant as a random factor) to that of a reduced model (i.e., the final model without condition as a factor in the model). The alpha level was set at 0.05. For effect sizes, partial Eta-squared ($\eta^2_p$) for each model was estimated using the "effectsize"[114] package in R. As a supplement to the null-hypothesis testing approach taken, we also ran Bayes factor analyses using the "BayesFactor"[115] package in R. Unlike $p$-values, where a non-significant effect cannot be interpreted as support for the null-hypothesis, Bayes Factors ($BF_{10}$) estimate the strength of evidence in favor of the alternative hypothesis (e.g., a $BF_{10}$ of 10 would suggest that the evidence in favor of the alternative hypothesis is ten times more likely than the null hypothesis, whereas a $BF_{10}$ of 0.10 would suggest that the evidence in favor of the null hypothesis is ten times more likely than the alternative hypothesis).

Simple linear regression models were then generated to determine: (i) whether groups differed on the dependent variable of interest, as well as any other dependent variables that could help with interpretation of the results (i.e., reaction times, accuracy and reversals); (ii) whether there was an effect of age on any of the dependent variables; and (iii) whether the effect of age on the dependent variables differed between groups (i.e., an age × group interaction effect). Thus, for these linear regression models, age, group, and their interaction were included as independent variables in the model. Where there was a significant main effect of group and an absence of a group interaction effect, two-tailed post hoc comparisons were performed using Tukey's HSD test. To supplement interpretation of the post hoc comparisons, Cohen's $d$ was estimated for each contrast. Where an interaction effect was present, simple slope analyses were conducted. The alpha level was set at 0.05 and $p$-values were adjusted for comparing a family of four estimates (adjusted $p$-values are denoted: $p_{Tukey}$).

Following group comparisons, we conducted a series of correlation analyses at the sample-level to determine whether tactile sensitivity, as assessed via the vibrotactile battery, would correlate with questionnaire-based measures of sensory and related social functioning. Pearson's correlation analyses were conducted between relevant tactile sensitivity metrics and scale items of the Sensory Processing Measure and Sensory Experience Questionnaire. Note that total scores (which are age-adjusted) were used for the Sensory Processing Measure, but raw scores (not age-adjusted) were used for the Sensory Experience Questionnaire. While the Sensory Processing Measure provides age-adjusted values, the Sensory Experience Questionnaire does not. Given the large number of correlation analyses being conducted here, the more conservative Bonferroni method was used to adjust the $p$-values to reduce Type 1 Error. All $p$-values were adjusted by multiplying the resulting p-value of each correlation between a given tactile sensitivity metric by

the number of subscale items within each questionnaire. For example, for all of the correlations conducted between static detection thresholds and the items of the Sensory Processing Measure, the resulting *p*-value was multiplied by 7 (i.e., number of subscale items within the Sensory Processing Measure).

Finally, we were interested in whether individual differences in tactile sensitivity for a were related to their disorder-specific clinical symptoms. Here, we conducted correlational analyses between the tactile sensitivity metrics to the domain or scales for each disorder-specific questionnaire (i.e., the Autism Diagnostic Observation Schedule, and Autism Diagnostic Interview-Revised for Autism-related symptomatology and the Conners, Conners-3, and DuPaul's ADHD rating scales for ADHD-related symptomatology). Again, Bonferroni adjustments of the *p*-values for each analysis was applied to control for the increase in Type I error, for each subscale. If readers are interested, group comparisons made on the Sensory Processing Measure and Sensory Experience Questionnaire can be found in Supplementary Results, Supplementary Table 2.

**Reporting summary**. Further information on research design is available in the Nature Research Reporting Summary linked to this article.

## Data availability
The raw data used in this study, which can be analyzed by the code provided in the links above is available on the Open Science Framework: https://osf.io/9mdc6/.

## Code availability
The code used to process the vibrotactile data collected from the Cortical Metrics device are available from https://github.com/HeJasonL/BATD. The code used to analyze the data and generate the figures in this study are also available as an RMarkdown[116] file on the Open Science Framework: https://osf.io/9mdc6/

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

## Acknowledgements

We would like to thank all the participants and their families who gave up their valuable time to help provide us with the data used in this study. We are also grateful to all the research co-ordinators and assistants that helped collect these data at various stages.

## Author contributions

J.L.H. performed data processing and analysis. J.L.H. wrote the manuscript with N.A.J.P. E.W., M.T., M.M., R.A.E. and S.H.M. provided comments on the manuscript and contributed to initial protocol and overall study design. N.A.J.P. contributed to research design and supervised all aspects of the manuscript, including data processing, analysis, and write up.

## Competing interests

The authors declare the following competing interests: This work was funded by NIH/NIMH R21MH098228, R01MH106564, R01MH078160, and R00MH107719. J.L.H. and N.A.P. received salary support from the Nancy Lurie Marks Family Foundation as part Autism Sensory Research Consortium. M.T. is the president and founder of Cortical Metrics, the company that developed the tactile stimulator used for this study. S.H.M. receives royalties for US Patent 10,410,041.
