## [Peer Review File · Communications Biology]

Reviewers' comments:

Reviewer #1 (Remarks to the Author):

He and colleagues show that alterations of tactile processing are disorder-specific and related to the core symptoms of ASD and ADHD. This paper is clearly of general interest for researchers in perception and cognition in ASD and/or ADHD.

The study presents the results of original research and the scientific quality and technical soundness of the work is excellent. The experiments are performed to a high technical standard and are described in clear and sufficient detail.

I appreciate the large sample for the experiments and the fact that the code used to process and analyse the data has been made available (together with their custom R package), with the promise of also making the data available once the paper is accepted.

I also appreciate the ability of authors to link their results to fundamental and generalised neurophysiological functions (such as abnormal GABA mediated neurotransmission) and to previous literature in the field.

1. My only major concern is with regards to the statistical analyses. The authors over-rely on p-values throughout the manuscript. For example: what's a "non-significant trend towards a main effect"; can't this be instead support for a null effect? Fortunately, it is possible to quantify the support for the alternative hypothesis compared to the null hypothesis using Bayesian statistics. That is: it is possible to ask how likely is the alternative hypothesis to have generated the data compared to the null hypothesis? Note that this method does not rely on arbitrary cutoffs. Software like JASP allows to quickly perform all the analyses reported in the manuscript in Bayesian form; alternatively, it is possible to use R code for this and the authors seem to be proficient in using R. This not only allows to disambiguate "non-significant trend towards a main effect" but, crucially, would also increase the validity of the main results which are often based on extremely variable data (Figure 2 to 5 and figure 7 often seem to show 'small' group differences when raw data are plotted) and p-values near the (arbitrary and absurd) .05 cutoff (see Benjamin, D. J., Berger, J. O., Johannesson, M., Nosek, B. A., Wagenmakers, E. J., Berk, R., ... & Cesarini, D. (2018). Redefine statistical significance. *Nature Human Behaviour*, 2(1), 6-10.). If at least the main conclusions survive the tests of Bayesian reanalyses, I am happy to recommend this paper for publication. Authors can report those additional analyses either in the text or as a separate supplementary material. Alternatively, report 95% confidence intervals and effect sizes.

2. Why for some statistics you report Bonferroni and for other Tukey corrections?

3. I think that the discussion would benefit by including a discussion of how the results link to other perceptual domains: do the disorder-specific alterations of detection, discrimination and order judgement in ASD and ADHD resemble similar results in other perceptual domains, such as visual or auditory perception?

Very minor:

4. Although the methods are reported in detail at the end of the experiment, before the presentation of results authors could report briefly some details of the tasks performed, in order to facilitate the flow of the manuscript.

5. Correlation of .28 became correlation of 28 (line 372)

6. Report statistics coherently; either use notation 0.XXX or notation .XXX

Reviewer #2 (Remarks to the Author):

This study investigates tactile sensitivity in neurotypical children, children with ASD, ADHD, and both. Multiple measures are used such as absolute thresholds, amplitude and frequency discrimination thresholds, and order judgements. The study reports different effects for the different groups that were also task-dependent. That is, children with ASD struggled more on some tasks, while other tasks elicited decreased performance in children with ADHD. Notably, some previous results in the literature might have been misattributed to ASD, while appearing more driven by ADHD, which might not have been tested for.

The paper presents an important contribution to the literature. The relatively high number of participants, spread over several groups, along with the focus on several measures of tactile sensitivity, rather than just a single one, make this a well designed and insightful study. The tactile tasks themselves are well established, which lends further credence to the results. I have no major concerns, however a number of relatively minor points that the authors might want to consider.

1) Regarding the specific tactile sensitivity measures, perhaps it is worth reminding the reader that these were chosen based on a presumed contribution of cortical processing, which is thought to be affected in these disorders. For example, another common measure of tactile sensitivity is spatial acuity, however this is mostly thought to be linked to peripheral mechanisms and specifically innervation density, and therefore unlikely to be affected. Perhaps this is obvious, though it might not be for every reader

2) It is great to see that a range of tactile sensitivity measures were employed in this study. However, it might be worth reminding the reader that these aren't exhaustive and that different cortical mechanisms might underlie even very similar tasks with different parameters. For example, the vibrotactile stimuli employed in the current study were all in the flutter range. It is not clear whether the results would translate to the (higher frequency) vibration range, as cortical studies have shown some different processing of Pacinian high-frequency input, for example.

3) The results from the amplitude and frequency discrimination tasks are very intriguing. However, I take issue with the statement that "it would be correct to say elevated discrimination thresholds are more common to children with an ASD diagnosis". First, while the mean thresholds differed across groups, this does not imply that every child in the ASD+ADHD group had elevated thresholds while the other kids did not. Second, even if this was the case, whether the statement above is generally true would depend on the prevalence of ADHD in children diagnosed with ASD, and it is not clear whether the reported sample in this study reflects the true frequency of co-diagnosis. This point should be clarified by the authors.

4) In how far can significant results be attributed to higher sample sizes in some groups than others? For example on the detection task the ASD group appears to exhibit a higher mean than the ADHD group, yet the latter is found to be significant, while the former is not, likely due to the difference in

group numbers.

5) To what extent do the tactile sensitivity measures reflect common cortical processing mechanisms? For example, the paper reports broadly similar findings for amplitude and frequency discrimination. How correlated are the thresholds from both tasks? Do children with higher amplitude thresholds generally also display higher frequency thresholds (independent of their group membership, for example analysis restricted to neurotypical children where sample size is the largest)? This might further bolster the notion that both types of tasks rely on at least partially overlapping cortical circuits.

6) The number of participants is unclear, as different numbers appear in different sections of the paper. For example, in the introduction it is stated that the ASD group included 45 children, while the Methods section states that this number was 34.

7) I appreciate that the authors did not want to include the results of every test they ran in the main text. However, it might still be helpful to provide a list of all tasks that were run so that readers do not have to refer to the supplementary material simply to find out the different tasks.

Reviewer #3 (Remarks to the Author):

The manuscript entitled “Disorder-specific alterations of tactile sensitivity in neurodevelopmental disorders” describes the (i) assessment of tactile perception in ASD, ADHD (detection, discrimination and TOJ) ASD + ADHD groups and (ii) whether perceptual profiles in each group are correlated with higher-level, characteristic symptoms in each groups. Disorder- specific vs -general perceptual characteristics was assessed; results suggested detection & TOJ difficulties more common to ADHD, while discrimination was more common for the ASD group. Tactile profiles also correlated to higher-level symptoms in each group.

Abstract :

“Strikingly, subsequent correlation analyses found that the disorder-specific alterations suggested by the group comparisons were also exclusively related to the core symptoms of each respective disorder. “ – sentence could be more clearly stated (maybe remove “strikingly”)

Introduction:

The use of “touch” exclusively in the first paragraph of the manuscript reads awkward – suggest “tactile” throughout ..

P 3 – “In the context of sensory processing abnormalities in ASD and ADHD, it is possible some or even all of the previously identified sensory abnormalities in one disorder (e.g., elevated detection thresholds for sensory stimuli in ASD) were simply due to the pathophysiology of the other (i.e., ADHD). It is also possible that sensory abnormalities that would not otherwise be present in isolated cases of either disorder are only present in individuals with a co-diagnosis of ASD + ADHD. Indeed, having ASD + ADHD could even amplify any existing sensory abnormalities specific to each disorder in isolation.” - ok. However, need to provide evidence to argue this – ie, describe evidence of hyper/hypo sensitivity in ADHD, etc. Also need to be sure that previous studies demonstrating hyper/hypo sensitivity in ASD assessed did not assess ADHD symptomatology , - did any do this ? In

what way would having ADHD “amplify” sensory abnormalities in ASD? Do the authors mean in magnitude – i.e., more hyper/hypo sensitive? Do the authors mean in frequency - i.e., more often hyper/hypo sensitive? By abnormal, do the authors mean hypersensitive or hyposensitive, or both? Does the directionality of the abnormality matter – why / why not? There is a very large literature in other modalities (i.e., visual and auditory abnormalities published in ASD (ADHD as well) re spatial & temporal frequency, MSI, etc.) that is not acknowledged here that could be used champion the statements above, and used to have a much interesting and informative interpretation of the results.

This is especially relevant here since in the discussion, the authors suggest that ; “ ... the disorder-specific alterations of tactile processing we identified with detection, discrimination and order judgement in ASD and ADHD are likely to reflect a dysfunction of more fundamental and generalized neurophysiological functions required for processing across sensory domains (i.e., they are domain-general) rather than a dysfunction of sensory processing specific to the tactile domain (i.e., domain-specific alterations).- p25”

These are important statements that need to be supported, especially given tactile results (usually poorer performance for ASD+ADHD group). Pls clarify.

Results

I strongly suggest effect sizes be included for all tactile performance analysis in order to have a better understanding of the group differences - are these group differences substantive?

The authors report accuracy and RT measures (often significant group differences) but do not really discuss whether Accuracy, for example, affects performance, or if increased Accuracy is consistently specific to one condition over another (seems to be the case of ASD+ADHD group).

“Non-significant trends” difficult to interpret without effect sizes.

This is a significant amount of discussion/interpretation within the results section that can be moved on to the discussion = redundant and repetitive text. i.e., p14 line 246 through p15 line 257, for example ...)

P 14 “ difficulties with tactile discrimination appear to be more common to children with ASD + ADHD. “ –

P 14 – “While our results appear to run counter to the findings of those studies, it is again possible they had not accounted for ADHD comorbidity.” – A more nuanced interpretation is needed here/discussion; can use this argument to argue several other potential non-differences found between ASD and TDC groups. This stems from the lack of lit review describing how sensory abnormalities have been explained in studies assessing sensory domains in intro – ie, age of groups assessed, methods, physiological differences between modalities, theories so of atypical sensory abilities in ASD or ADHD, etc.

– However, it is worth considering that although the ASD group did not significantly differ from the TDC group on comparisons of the discrimination thresholds, given the majority of children in our sample with ASD also had a co-diagnosis of ADHD (~64%), it would be correct to say elevated discrimination thresholds are more common to children with an ASD diagnosis.” It is correct to say this but is it really interpretable? This statement actually detracts from the potential significance/importance of the findings and I strongly suggest it be removed (how can you interpret 64 % ???).

P 16 - "wider temporal binding windows' have also been found in ASD but studies not cited/discussed = implications on specificity of finding.

P18 - "Second, detection thresholds being associated with both hyper- and hyposensitivity is interesting in that this would appear to run counter to the common narratives of hyper- and hypo-responsivity as either distinct traits or opposite extremes along a single axis." This same view/argument/discussion re non-distinct hyper- and hyposensitivity for perceptual findings would benefit the manuscript greatly. See comments above

P 18 - "Correlation analyses between tactile sensitivity thresholds and subscale items from the SPM and SEQ : After adjusting for multiple comparisons, ..." - pls explain how many correlations were assessed / how were p-values adjusted? Not clear.

P22 - line 388 - 2 long, running sentence.. pls edit.

Results general - way too much text regarding interpretation is included in the Results section, that could be presented in the discussion (can use subheadings to organize, etc ...)

Discussion.

The Introduction and Discussion sections read very differently - through they were written by different authors - whereby the GABA heavy interpretation in the Discussion is not contextualized in the Introduction as a potential physiological explanation. The disconnect between Intro and Discussion sections is in part due, again, to the fact that a very large literature in other modalities - some of which suggest altered GABA-ergic transmission as a potential substrate for atypical sensory processes (in ASD at least) - is not presented. This disconnect between section takes away from the manuscript / importance of research question/findings.

Based on:

- high detection thresholds/TOJ more commonly elevated in those with a diagnosis of ADHD (primary or secondary diagnosis) and positively scaled with hyperactivity/inattention (specific to ADHD)

- high discrimination thresholds more commonly elevated in those with a diagnosis of ASD and scaled specifically with communication scores on the ADOS (specific to ADOS)

Lower-level sensory alteration have suggested altered-GABA transmission (and consequential neural alterations affecting early perception) in ASD, and could explain altered discrimination threshold (as has been posited in visual domain but not discussed). However, the link to higher-level ASD behaviours is minimal "While speculative, it is possible that alterations GABAergic functioning, which may explain domain-general alterations of sensory discrimination, in turn contributes to many of the higher-order behavioral symptoms of ASD." - Given the emphasis put GABA-ergic dysfunction, a more neurophysiologically -based explanation - even if speculative - would be more appreciated. For example, how does/do local alterations resulting from GABA-ergic alterations possible effect the development of larger neural networks mediating perceptual processes closer to social behavior in autism. ie, face perception, biological motion, etc. How do these early perceptual abnormalities cascade to higher-level cognition/ behavior? Or Is altered lateral connectivity - as suggested by tactile results - also define altered connectivity in higher-order brain regions that are closer to behavior. These type of arguments would be more interesting/informative.

However, the explanation to how altered GABA-ergic transmission result in increased detection/TOJ threshold sin ADHD is less evident (maybe impossible) and less convincing as presenting (i.e., relationship w Tourette's, time perception but not TOJ, etc.). If the authors are set on the GABA

dysfunction interpretation, then this is very difficult (common to detection and discrimination). Maybe interpret results in terms of broader neurophysiological context, such as GABA-related excitation/inhibition balance ... just a suggestion.

Methods

Psychophysical approach solid

Much smaller group with ASD only vs ADHD only. More importantly, it is not clear in the Methods how participants were recruited - from a database? Did they already have a Dx – or were participants with ASD then assessed for ADHD .. pls clarify.

Covarying for non-verbal IQ acceptable.

General response to reviewer comments

We appreciate the opportunity to revise our manuscript and thank all reviewers for their considerate comments. We have carefully considered the comments by each of the reviewers, have made changes to the manuscript, and have provided responses to each comment below. Each reviewer comment is in bold, followed by our response. We have made a stronger effort to contextualize our findings using the results of prior studies (based on comments by reviewer 3). This involved adding sections to the Introduction, as well as elaborating on points made in the Discussion. As highlighted in the Response, all the reviewers made comments regarding the statistics (i.e., over-reliance on p-values as per reviewer 1, sample size as per reviewer 2 and effect sizes by reviewer 3). In response to those comments, we have included Bayes factors for each of our analyses, as well as effect sizes for each of our comparisons. These additions bolster our interpretation of the findings. We strongly believe that the comments provided by the reviewers in this review process has greatly improved the impact of our manuscript. We thank the editor and reviewers for their time and effort.

Reviewer #1 (Remarks to the Author):

He and colleagues show that alterations of tactile processing are disorder-specific and related to the core symptoms of ASD and ADHD. This paper is clearly of general interest for researchers in perception and cognition in ASD and/or ADHD. The study presents the results of original research and the scientific quality and technical soundness of the work is excellent. The experiments are performed to a high technical standard and are described in clear and sufficient detail. I appreciate the large sample for the experiments and the fact that the code used to process and analyse the data has been made available (together with their custom R package), with the promise of also making the data available once the paper is accepted. I also appreciate the ability of authors to link their results to fundamental and generalised neurophysiological functions (such as abnormal GABA mediated neurotransmission) and to previous literature in the field.

R1.1. My only major concern is with regards to the statistical analyses. The authors over-rely on p-values throughout the manuscript. For example: what's a "non-significant trend towards a main effect"; can't this be instead support for a null effect? Fortunately, it is possible to quantify the support for the alternative hypothesis compared to the null hypothesis using Bayesian statistics. That is: it is possible to ask how likely is the alternative hypothesis to have generated the data compared to the null hypothesis? Note that this method does not rely on arbitrary cutoffs. Software like JASP allows to quickly perform all the analyses reported in the manuscript in Bayesian form; alternatively, it is possible to use R code for this and the authors seem to be proficient in using R. This not only allows to disambiguate "non-significant trend towards a main effect" but, crucially,

would also increase the validity of the main results which are often based on extremely variable data (Figure 2 to 5 and figure 7 often seem to show 'small' group differences when raw data are plotted) and p-values near the (arbitrary and absurd) .05 cutoff (see Benjamin, D. J., Berger, J. O., Johannesson, M., Nosek, B. A., Wagenmakers, E. J., Berk, R., ... & Cesarini, D. (2018). Redefine statistical significance. *Nature Human Behaviour*, 2(1), 6-10.). If at least the main conclusions survive the tests of Bayesian reanalyses, I am happy to recommend this paper for publication. Authors can report those additional analyses either in the text or as a separate supplementary material. Alternatively, report 95% confidence intervals and effect sizes.

We agree with the sentiment of the over-reliance on p-values in our manuscript. Admittedly, we stuck to the traditional means of analyzing and interpreting the data (through p-values) to increase the interpretability of the paper for the majority of readers (which we believed would include clinicians, who may or may not be as familiar with Bayesian statistics as those in clinical research). However, upon consideration of the reviewer's comment, we recognize that others may be familiar with non-frequentist approaches (i.e., Bayesian) adopting Bayesian approaches as standard practice. We have now included the Bayesian statistics for each of our analyses as suggested by the reviewers and have included them in the main body of our manuscript where possible. While we were familiar with JASP, for the purpose of consistency with our other analysis we conducted the analyses in R so that we could include these analyses in our RMarkdown file (which has also been updated on the OSF to include the Bayesian analyses, and additional effect sizes and confidence intervals).

As you will be able to see in the revised manuscript, the main conclusions remain despite the reanalysis.

R1.2. Why for some statistics you report Bonferroni and for other Tukey corrections?

We have a preference for using Tukey's correction for multiple comparisons for post-hoc testing due to the belief that Tukey's provides a better balance of reducing Type-1 and Type-2 errors compared to the Bonferroni approach, which we believe is too conservative and can result in type-2 errors. For the correlation analyses, however, we switched to the use of Bonferroni's for correction as we conducted a substantial number of analyses and aimed to be as stringent as possible as to not overinterpret the results. See page 39, lines 19 to 20.

R1.3. I think that the discussion would benefit by including a discussion of how the results link to other perceptual domains: do the disorder-specific alterations of detection, discrimination and order judgement in ASD and ADHD resemble similar results in other perceptual domains, such as visual or auditory perception?

While we briefly discussed how our results tie into findings from other perceptual domains in the first paragraph of the discussion section, we recognize that this section could have been better expanded upon. We have since extended the discussion of how our results link across perceptual domains and have included more references to earlier work. See page 27, lines 9 to 10.

Very minor

R1.4. Although the methods are reported in detail at the end of the experiment, before the

presentation of results authors could report briefly some details of the tasks performed, in order to facilitate the flow of the manuscript.

A brief description of the tasks is provided in Figure 1. Given that the figure will be presented at the start of the results section, readers should be able to read over the figure caption before going on to read our results. We have revised the opening paragraph of the results section to more explicitly point the reader to the figure caption which contains sufficient detail for the reader. See page 6, line 13.

R1.5. Correlation of .28 became correlation of 28 (line 372)

We thank the reviewer for identifying this error and have fixed this error in the revised manuscript. See page 22, line 6.

R1.6. Report statistics coherently; either use notation 0.XXX or notation .XXX

We apologize for this oversight. We have since standardized the notation to 0.XXX.

Reviewer #2 (Remarks to the Author):

This study investigates tactile sensitivity in neurotypical children, children with ASD, ADHD, and both. Multiple measures are used such as absolute thresholds, amplitude and frequency discrimination thresholds, and order judgements. The study reports different effects for the different groups that were also task-dependent. That is, children with ASD struggled more on some tasks, while other tasks elicited decreased performance in children with ADHD. Notably, some previous results in the literature might have been misattributed to ASD, while appearing more driven by ADHD, which might not have been tested for. The paper presents an important contribution to the literature. The relatively high number of participants, spread over several groups, along with the focus on several measures of tactile sensitivity, rather than just a single one, make this a well designed and insightful study. The tactile tasks themselves are well established, which lends further credence to the results. I have no major concerns, however a number of relatively minor points that the authors might want to consider.

R2.1. Regarding the specific tactile sensitivity measures, perhaps it is worth reminding the reader that these were chosen based on a presumed contribution of cortical processing, which is thought to be affected in these disorders. For example, another common measure of tactile sensitivity is spatial acuity, however this is mostly thought to be linked to peripheral mechanisms and specifically innervation density, and therefore unlikely to be affected. Perhaps this is obvious, though it might not be for every reader

We thank the reviewer for this suggestion. We have now amended the Introduction to highlight why these tasks were selected (i.e., due to their ability to indirectly assess individual differences in cortical processing). See page 6, lines 12 to 16. Spatial processing is indeed also of strong interest but not possible to examine using the stimulator used here. One could argue abnormal spatial processing could be due to cortical mechanisms as well as peripheral mechanisms (e.g. lateral inhibition likely encodes spatial information).

R2.2. It is great to see that a range of tactile sensitivity measures were employed in this study. However, it might be worth reminding the reader that these aren't exhaustive and that different cortical mechanisms might underlie even very similar tasks with different parameters. For example, the vibrotactile stimuli employed in the current study were all in the flutter range. It is not clear whether the results would translate to the (higher frequency) vibration range, as cortical studies have shown some different processing of Pacinian high-frequency input, for example.

We agree with the reviewer and have now highlighted that the measures we assessed are not exhaustive and that different cortical mechanisms may underlie similar tasks with slightly different parameters (i.e., our results might not hold across at higher stimulation frequencies). In fact, the literature in ASD suggests differences between flutter and vibration, and we reviewed this in our prior review (Mikkelsen et al.). See page 29, lines 19 to 23, and page 30, lines 1 to 10.

R2.3. The results from the amplitude and frequency discrimination tasks are very intriguing. However, I take issue with the statement that “it would be correct to say

elevated discrimination thresholds are more common to children with an ASD diagnosis”.

First, while the mean thresholds differed across groups, this does not imply that every child in the ASD+ADHD group had elevated thresholds while the other kids did not. Second, even if this was the case, whether the statement above is generally true would depend on the prevalence of ADHD in children diagnosed with ASD, and it is not clear whether the reported sample in this study reflects the true frequency of co-diagnosis. This point should be clarified by the authors.

We strongly agree with the points touched on by this particular comment. In fact, we had originally included a whole paragraph discussing how overlap between groups should be considered when interpreting the data. Due to the word count, we included that paragraph in supplementary materials as opposed to including it in the main body of the manuscript. We recognize that the we did not bring sufficient attention to the existence of this paragraph in the main manuscript. For this reason, we have now updated the Discussion section to a) highlight these limitations as factors to consider and b) better direct readers to the supplementary materials where we discuss these limitations in more detail. See page 29, line 22.

As for the second point (statement being dependent on whether the prevalence of ASD + ADHD in our sample is the true frequency of co-diagnosis), we thank the reviewer for highlighting this. We have removed the quoted statement from the manuscript. We have also included a brief discussion about whether the frequency of ADHD comorbidity in our sample is reflective of the population frequency of comorbidity. Page 31, lines 9 to 13.

R2.4. In how far can significant results be attributed to higher sample sizes in some groups than others? For example on the detection task the ASD group appears to exhibit a higher mean than the ADHD group, yet the latter is found to be significant, while the former is not, likely due to the difference in group numbers.

We recognize this issue. For this reason (and for the reasons suggested by other reviewers), we have now included effect size and Bayes factors for each of our analyses to help better contextualize these findings for interpretation. For instance, for static detection, the addition of the effect size suggests that the difference between ADHD and controls had a notably larger effect size (0.70) than the difference between ASD and controls (0.34). Similarly, the Bayes factors for the post-hoc tests show strong evidence in favor of there being a group difference with regard to the comparison between ADHD and controls ($BF_{10} = 2803.56$ – 2803 times more likely the alternative), the Bayes factors show weak evidence *against* there being a difference between ASD and controls ($BF_{10} = 0.85$ == $BF_{01} = 1.17$ – i.e., evidence for ASD > controls is 1.17 times more likely).

R2.5. To what extent do the tactile sensitivity measures reflect common cortical processing mechanisms? For example, the paper reports broadly similar findings for amplitude and frequency discrimination. How correlated are the thresholds from both tasks? Do children with higher amplitude thresholds generally also display higher frequency thresholds (independent of their group membership, for example analysis restricted to neurotypical

children where sample size is the largest)? This might further bolster the notion that both types of tasks rely on at least partially overlapping cortical circuits.

We have run the correlation between amplitude and frequency discrimination thresholds. As can be discerned, these two measures correlate across groups. See figure and figure caption below.

We have also now included a brief sentence discussing how the sensitivity measures may reflect common cortical processing mechanisms and have provided the figure in our response in the revised supplementary materials as well. Page 15, lines. 6 to 8.

Supplementary Fig. 12. Correlations between amplitude discrimination and sequential frequency discrimination. Plots are shown at the total sample level (a) and in controls only (b), ASD, (c) ASD + ADHD and

(d) ADHD. While some correlations are not significant, this is likely due to power, since the correlation at the total sample level is significant, with an effect that is comparable to those seen at the independent groups level. ADT = Amplitude Discrimination Threshold; SMFD = Simultaneous Frequency Discrimination

R2.6. The number of participants is unclear, as different numbers appear in different sections of the paper. For example, in the introduction it is stated that the ASD group included 45 children, while the Methods section states that this number was 34.

We thank the reviewer for identifying this. The correct numbers were those presented in the methods section. We have now fixed the values in the introduction. See page 6, lines 9 to 11.

R2.7. I appreciate that the authors did not want to include the results of every test they ran in the main text. However, it might still be helpful to provide a list of all tasks that were run so that readers do not have to refer to the supplementary material simply to find out the different tasks.

We have now included a list of all the tasks in the figure caption of Figure 1 of the revised manuscript. See page 10, lines 20 to 22.

Reviewer #3 (Remarks to the Author):

The manuscript entitled “Disorder-specific alterations of tactile sensitivity in neurodevelopmental disorders” describes the (i) assessment of tactile perception in ASD, ADHD (detection, discrimination and TOJ) ASD + ADHD groups and (ii) whether perceptual profiles in each group are correlated with higher-level, characteristic symptoms in each groups. Disorder- specific vs -general perceptual characteristics was assessed; results suggested detection & TOJ difficulties more common to ADHD, while discrimination was more common for the ASD group. Tactile profiles also correlated to higher-level symptoms in each group.

Abstract

R3.1. *“Strikingly, subsequent correlation analyses found that the disorder-specific alterations suggested by the group comparisons were also exclusively related to the core symptoms of each respective disorder” – sentence could be more clearly stated (maybe remove “strikingly”)*

We have removed the term strikingly and have restated the sentence for clarity. See page 3, lines 10 to 13.

Introduction

R3.2. The use of “touch” exclusively in the first paragraph of the manuscript reads awkward – suggest “tactile” throughout

We thank the reviewer for highlighting this. We have removed the majority of instances in which the word ‘touch’ is used and have replaced it with ‘tactile processing’. We have retained the use of the word ‘touch’ when it is being referred to as a sense (e.g., “touch, smell and taste) and when it is being referred to in the context of ‘social touch’. See page 1, lines 2, 7 and 13.

R3.3. P3 - *“In the context of sensory processing abnormalities in ASD and ADHD, it is possible some or even all of the previously identified sensory abnormalities in one disorder (e.g., elevated detection thresholds for sensory stimuli in ASD) were simply due to the pathophysiology of the other (i.e., ADHD). It is also possible that sensory abnormalities that would not otherwise be present in isolated cases of either disorder are only present in individuals with a co-diagnosis of ASD + ADHD. Indeed, having ASD + ADHD could even amplify any existing sensory abnormalities specific to each disorder in isolation.”* - **ok. However, need to provide evidence to argue this – ie, describe evidence of hyper/hypo sensitivity in ADHD, etc. Also need to be sure that previous studies demonstrating hyper/hypo sensitivity in ASD assessed did not assess ADHD symptomatology - did any do this?**

We have clarified our argument and added more references. See page 5, lines 4 to 13. Indeed, none of these studies had reported having assessed ADHD symptomatology.

R3.4. In what way would having ADHD “amplify” sensory abnormalities in ASD? Do the authors mean in magnitude – i.e., more hyper/hypo sensitive? Do the authors mean in frequency - i.e., more often hyper/hypo sensitive? By abnormal, do the authors mean hypersensitive or hyposensitive, or both? Does the directionality of the abnormality matter

– why / why not? There is a very large literature in other modalities (i.e., visual and auditory abnormalities published in ASD (ADHD as well) re spatial & temporal frequency, MSI, etc.) that is not acknowledged here that could be used champion the statements above, and used to have a much interesting and informative interpretation of the results. This is especially relevant here since in the discussion, the authors suggest that ; “ the disorder-specific alterations of tactile processing we identified with detection, discrimination and order judgement in ASD and ADHD are likely to reflect a dysfunction of more fundamental and generalized neurophysiological functions required for processing across sensory domains (i.e., they are domain-general) rather than a dysfunction of sensory processing specific to the tactile domain (i.e., domain-specific alterations).- p25” These are important statements that need to be supported, especially given tactile results (usually poorer performance for ASD+ADHD group). pls clarify.

The statement where we suggest that having ASD + ADHD could amplify sensory abnormalities in ASD was intended to be general (readers would need to have seen the results before a more specific discussion could be put forward). We have replaced ‘amplify’ to ‘mediate or moderate’ to account for the fact that the changes could be in either direction (as rightly pointed out by the reviewer). Similarly, we have removed the phrase ‘sensory abnormalities’ and replaced it with ‘sensory sensitivities’. We believe that “sensitivities” is more accurate/specific and can refer to both hypo- and hyper-sensitivities as our findings reveal most children show both. See page 5, lines 12 to 13.

We thank the reviewer for their suggestion and have included more references to the statements where we discuss how sensory sensitivities identified in ASD have also been identified in ADHD, and we agree with the reviewer that this strengthens the argument. See page 5, lines 4 to 13.

Results

R3.5. I strongly suggest effect sized be included for all tactile performance analysis in order to have a better understanding of the group differences - are these group differences substantive?

We agree that effect sizes would help with the interpretability of our results and have now included effect sizes (as well as Bayes factors – due to the recommendation of another reviewer, R1.1.) in the revised manuscript.

R3.6. The authors report accuracy and RT measures (often significant group differences) but to not really discuss whether Accuracy, for example, effects performance, or if increased Accuracy is consistently specific to one condition over another (seems to be the case of ASD+ADHD group). “Non-significant trends” difficult to interpret without effect sizes.

With regard to RTs and accuracy, they are intended to be used as supplemental information to thresholds. We avoided attributing too much meaning to RTs and accuracy since it is difficult to

separate these measures from threshold. For example, a person who has difficulty with detecting a stimulus is also going to be less accurate (indeed, a lower threshold necessitates that participant having been less accurate). In a similar vein, it is difficult to make inferences about RTs since longer RTs could suggest difficulties with perception (perhaps due to noise, which is interesting in itself and has been discussed by e.g. Ward, 2019), but could also simply reflect longer decision times (which we would expect to be longer if they are having more difficulty completing the task). Due to these reasons, we are reticent about making inferences based on these data and therefore only report these in Supplementary materials. As for the non-significant trends, we have now added effect sizes and Bayes factors to all of our statistics to aid interpretation.

- Ward, J. (2019). Individual differences in sensory sensitivity: A synthesizing framework and evidence from normal variation and developmental conditions. *Cognitive neuroscience*, 10(3), 139-157.

R3.7. There is a significant amount of discussion/interpretation within the results section that can be moved on the discussion = redundant and repetitive text. i.e., p14 line 246 through p15 line 257, for example ...) P 14 “difficulties with tactile discrimination appear to be more common to children with ASD + ADHD.

The discussion and interpretation included in the Results section is something that is not uncommon in studies previously published in the target journal and those like it (e.g., Communications Biology, Nature Human Behaviour and Nature Neuroscience). However, we

recognize that the brief discussion and interpretation offered in the results section did not allow for the more nuanced discussion and interpretation that is required for those statements. For this reason, we have made an effort to move some of the discussion and interpretation from the results into the discussion section, where there was space to elaborate. Examples: Page 14, page 18, and what was page 24 (moved out of results and into discussion).

R3.8. “– P 14 – “While our results appear to run counter to the findings of those studies, it is again possible they had not accounted for ADHD comorbidity.” – A more nuanced interpretation is needed here/discussion; can use this argument to argue several other potential non-differences found between ASD and TDC groups. This stems from the lack of lit review describing how sensory abnormalities have been explained in studies assessing sensory domains in intro – ie, age of groups assessed, methods, physiological differences between modalities, theories so of atypical sensory abilities in ASD or ADHD, etc.

The section being referred to here has been moved from the results section to the discussion section (based on comment 7 above). A more nuanced discussion about discrimination in ASD can now be found on page 26, lines 6 to 23, and page 27, lines 1 to 14.

R3.9. “*However, it is worth considering that although the ASD group did not significantly differ from the TDC group on comparisons of the discrimination thresholds, given the majority of children in our sample with ASD also had a co-diagnosis of ADHD (~64%), it would be correct to say elevated discrimination thresholds are more common to children with an ASD diagnosis.*” - It is correct to say this but is it really interpretable? This statement

actually detracts from the potential significance/importance of the findings and I strongly suggest it be removed (how can you interpret 64 % ???).

We have removed this section as suggested by the reviewer. See page 14, line 21. With regard to the high frequency of comorbidity, this was also a surprise to us during data collection. However, when we consulted the literature, it appears that this value is actually not too different from those reported in previous studies. We have now made reference to this point in the revised manuscript (see page 31, lines 9 to 13).

R3.10. P 16 - “wider temporal binding windows’ have also been found in ASD but studies not cited/discussed = implications on specificity of finding.

We have now added in the references and discussion on the implications of this particular finding. See page 16, lines 9 to 13.

R3.11. P18 – “Second, detection thresholds being associated with both hyper- and hyposensitivity is interesting in that this would appear to run counter to the common narratives of hyper- and hypo-responsivity as either distinct traits or opposite extremes along a single axis.” This same view/argument/discussion re non-distinct hyper- and hyposensitivity for perceptual findings would benefit the manuscript greatly. See comments above

We have now included a discussion of non-distinct hyper- and hyposensitivity across sensory domains. See page 29, lines 12 to 17.

R3.12. P 18 – “Correlation analyses between tactile sensitivity thresholds and subscale items from the SPM and SEQ : After adjusting for multiple comparisons, ...” – pls explain how many correlations were assessed / how were p-values adjusted? Not clear.

In our original submission, details of how multiple comparisons were adjusted for were described in the figure caption of Figure 6. We now recognize that this was not optimally placed. In the revised manuscript, we have clarified that the adjustment was done via the Bonferroni method and have explicitly referred readers to the section where further details are provided (i.e., the methods section). See page 17 lines 21 and page 22 lines 1 (for explicit reference to methods section). See further details now in methods section, page 39, lines 19 to 22, and page 40 lines 1 to 2.

R3.13. P22 – line 388 – 2 long, running sentence. pls edit.

Due to making the changes for comment R3.7, we have moved this section the beginning of the discussion, where we have made edits to this sentence. The paragraph as a whole has been shortened to keep the manuscript within a reasonable word limit. See page 24, lines 1 to 19.

R3.14. Results general – way too much text regarding interpretation is included in the Results section, that could be presented in the discussion (can use subheadings to organize, etc.,)

As per our response to comment R3.7, we have now moved most of the interpretation in the results to the discussion section. We have also included subheadings to help organize the discussion.

Discussion

R3.15. The Introduction and Discussion sections read very differently - through they were written by different authors – whereby the GABA heavy interpretation in the Discussion is not contextualized in the Introduction as a potential physiological explanation. The disconnect between Intro and Discussion sections is in part due, again, to the fact that a very large literature in other modalities – some of which suggest altered GABA-ergic transmission as a potential substrate for atypical sensory processes (in ASD at least) - is not presented. This disconnect between section takes away form the manuscript / importance of research question/findings.

We apologize for the disconnect between the writing style of the Introduction and Discussion. The reasoning for this was that the introduction was meant to function as a general introduction to sensory perception/ tactile processing in autism, while the Discussion was meant to be a place for providing a mechanistic explanation of the results. We attempted to bridge the focus of the

Introduction (general introduction) and Discussion (mechanistic explanations) with the comments made in our results section. We recognize that this was not sufficient and have made an effort to try to reduce this by preempting the discussion of GABA and its implications for sensory processing/perception in the Introduction. See page 6, lines 9 to 16 and page 7, lines 15 to 16.

R3.16. *“Lower-level sensory alteration have suggested altered-GABA transmission (and consequential neural alterations affecting early perception) in ASD, and could explain altered discrimination threshold (as has been posited in visual domain but not discussed). However, the link to higher-level ASD behaviours is minimal “While speculative, it is possible that alterations GABAergic functioning, which may explain domain-general alterations of sensory discrimination, in turn contributes to many of the higher-order behavioral symptoms of ASD.”*

– Given the emphasis put GABA-ergic dysfunction, a more neurophysiologically -based explanation - even if speculative - would be more appreciated. For example, how does/do local alterations resulting from GABA-ergic alterations possible effect the development of larger neural networks mediating perceptual processes closer to social behavior in autism. ie, face perception, biological motion, etc. How do these early perceptual abnormalities cascade to higher-level cognition/ behavior? Or Is altered lateral connectivity - as suggested by tactile results – also define altered connectivity in higher-order brain regions that are closer to behavior. These types of arguments would be more interesting/informative.

We thank the reviewer for their suggestion here. We had written but removed some of the neurophysiological explanations in prior iterations of our manuscript. We have now re-integrated

some of these explanations into the main body of the manuscript. See page 26, lines 6 to 23, and page 27, lines 1 to 14.

R3.17. However, the explanation to how altered GABA-ergic transmission result in increased detection/TOJ thresholds in ADHD is less evident (maybe impossible) and less convincing as presenting (i.e., relationship w Tourette's, time perception but not TOJ, etc.). If the authors are set on the GABA dysfunction interpretation, then this is very difficult (common to detection and discrimination). Maybe interpret results in terms of broader neurophysiological context, such as GABA-related excitation/inhibition balance ... just a suggestion.

We agree that relying solely on a GABA-ergic transmission explanation is not sufficient to explain the elevated detection thresholds and TOJ thresholds in ADHD. It is not immediately obvious how differences in GABA alone would lead to altered timing perception. We have now provided other possible neurophysiological explanations for elevated detection and TOJ thresholds in ADHD (i.e., with additional reference to the dopaminergic system and its implications on time processing in particular). See page 28, lines 8 to 18.

Methods

R3.18. Psychophysical approach solid

We thank the reviewer for this comment.

R3.19. Much smaller group with ASD only vs ADHD only. More importantly, it is not clear in the Methods how participants were recruited - from a database? Did they already have a Dx – or were participants with ASD then assessed for ADHD .. pls clarify. Participants were recruited from ...

We have now included a section discussing the higher frequency of ASD + ADHD (relative to ASD alone) in our sample (see page 31, lines 9 to 13). We have also now made the recruitment strategy clearer in the revised manuscript (page 31, lines 19 to 21). Indeed, while some children already had a co-diagnosis of ASD + ADHD, all children with ASD were also assessed for ADHD symptoms (using a combination of the DICA, K-SADS, Conner's and/or DuPaul ADHD rating scale). The information was then reviewed by a child neurologist or clinical psychologist to confirm diagnosis/diagnoses.

R3.20. Covarying for non-verbal IQ acceptable.

We thank the reviewer for this comment.

REVIEWERS' COMMENTS:

Reviewer #1 (Remarks to the Author):

The authors did a remarkable job. They have answered in detail all the issues raised by the reviewers. In particular, my comments regarding the statistical analyses - this was my only major concern - have been addressed fully and in depth. I appreciate the inclusion of Bayes Factor and effect sizes; these additional statistics strengthen the interpretation of the results. I recommend this paper for publication.

Reviewer #2 (Remarks to the Author):

All my concerns have been addressed and I recommend publication.

Reviewer #3 (Remarks to the Author):

Thank you for answering all the queries, addressing the comments, and editing the manuscript - I apologize fo the delay in reviewing.